



# Quantifying the Impacts of Atmospheric Rivers on the Surface Energy Budget of the Arctic Based on Reanalysis

Chen Zhang[1,2], John J. Cassano[1,2,3], Mark Seefeldt[1,2], Hailong Wang[4], Weiming Ma[4], Wen-wen Tung[5]

[1]Cooperative Institute for Research in Environmental Sciences, University of Colorado Boulder, Boulder, Colorado, USA
[2]National Snow and Ice Data Center, University of Colorado Boulder, Boulder, Colorado, USA
[3]Department of Atmospheric and Oceanic Sciences, University of Colorado Boulder, Boulder, Colorado, USA
[4]Atmospheric Sciences and Global Change Division, Pacific Northwest National Laboratory, Richland, Washington, USA
[5]Department of Earth, Atmospheric, and Planetary Sciences, Purdue University, West Lafayette, Indiana, USA

*Correspondence to*: Chen Zhang (chen.zhang-3@colorado.edu) and John J. Cassano (john.cassano@colorado.edu)

**Abstract.** We present a comprehensive analysis of Arctic surface energy budget (SEB) components during atmospheric river (AR) events identified by integrated water vapor transport exceeding the monthly 85th percentile climatological threshold in 3-hourly ERA5 reanalysis data from January 1980 to December 2015. Analysis of average anomalies in SEB components, net SEB, and the overall AR contribution to the total seasonal SEB reveals clear seasonality and distinct land - sea - sea ice contrast patterns. Over the sea ice-covered central Arctic Ocean, ARs significantly impact net SEB, inducing substantial surface
warming in fall, winter, and spring, primarily driven by large anomalies in surface downward longwave radiation. We find that ARs make a substantial relative contribution to the mean SEB in spring (32%), exceeding their corresponding occurrence frequency (11%). However, in other seasons, ARs contribute relatively less to the mean SEB than their frequency, indicating a diminished role compared to their occurrence frequency. Over sub-polar oceans, ARs have the most substantial positive impact on net SEB in cold seasons, mainly attributed to significant positive turbulent heat flux anomalies, with a maximum
contribution to the mean SEB in spring averaging 65%. In summer, ARs induce negative impacts on net SEB, primarily due to reduced shortwave radiation from increased cloud cover during AR events. Over continents, ARs generate smaller absolute impacts on net SEB but contribute significantly to the mean SEB in cold seasons, far surpassing their corresponding frequency, highlighting their crucial role in determining the net SEB over continents during cold seasons. Greenland, especially western Greenland, exhibits significant downward longwave radiation anomalies associated with ARs, which drive large net SEB
anomalies and contribute >54% to mean SEB, and induce amplified surface warming year-round. This holds significance for melt events, particularly during summer. This study quantifies the role of ARs on surface energy budget, contributing to our understanding of the Arctic warming and sea ice decline in ongoing Arctic amplification.



## 1 Introduction

The Arctic is a multifaceted environment, distinguished by close interactions among its atmosphere, ocean, sea ice and land
components, influenced by various forcing from lower latitudes, operating across a wide range of time and space scales
(Serreze et al., 2007). The Arctic has experienced disproportionate and accelerated warming compared to the global average
temperature increase over the past decades (Cohen et al., 2014; Graversen et al., 2008; Polyakov et al., 2002; Screen and
Simmonds, 2010; Serreze and Barry, 2011; Serreze et al., 2009; Serreze and Francis, 2006; Taylor et al., 2022). This
phenomenon is widely known as Arctic amplification and is a significant aspect of climate change in the Arctic.

Multiple physical mechanisms likely contribute to Arctic amplification. Locally, these mechanisms include the surface-albedo
feedback (Hall, 2004; Screen and Simmonds, 2010; Serreze et al., 2009; Serreze and Barry, 2011; Zhang et al., 2018), Planck
and lapse rate feedback (Pithan and Mauritsen, 2014; Zhang et al., 2018, 2021b), cloud-convection feedback and cloud-
radiative forcing (Abbot et al., 2009; Alexeev et al., 2005; Lee et al., 2017; Mortin et al., 2016; Shupe and Intrieri, 2004), along
with radiative effects associated with greenhouse gasses (Graversen and Wang, 2009; Shindell and Faluvegi, 2009). Moreover,
additional studies emphasize a remote perspective investigating strong poleward moisture and heat flux transports (Graversen
et al., 2008; Mortin et al., 2016; Park et al., 2015b), particularly through atmospheric rivers in recent years (Baggett et al.,
2016; Hegyi and Taylor, 2018; Mattingly et al., 2018, 2023, 2020; Neff, 2018), to explore Arctic amplification.

Atmospheric rivers (ARs) are long and narrow filaments of enhanced moisture transport typically associated with a low-level
jet and extratropical cyclone (Ralph et al., 2018). In recent years, there has been increasing research interest in the influence
of ARs on polar weather and climate. This growing attention is evident in various studies (Baggett et al., 2016; Gorodetskaya
et al., 2014; Guan et al., 2016; Ma et al., 2020, 2021; Mattingly et al., 2023, 2020; Wille et al., 2021; Zhang et al., 2023a, b;
Shields et al., 2022) that highlight the significance of ARs in enhancing moisture, downward infrared radiation, cloud-radiative
effects, precipitation patterns, and the surface energy balance. These complex interactions potentially contribute to sustained
surface warming in the Arctic region.

Over the Arctic, surface turbulent and radiative fluxes link the land, ocean and sea ice surface to the atmosphere. In general,
during the cold season, net heat fluxes transfer energy away from the surface to the atmosphere, facilitating the surface cooling
and winter sea ice formation and growth. These processes are mostly reversed, during the warm months, due to the strong net
solar radiation fluxes, facilitating surface warming and sea ice melting (Serreze et al., 2007). Previous studies have found that
the enhanced moisture transport associated with ARs leads to anomalously large downward longwave radiation and net
longwave radiation, which influences the subsequent surface radiation and energy budgets, resulting in enhanced surface
warming and sea ice decline over the Arctic in the boreal winter (Hegyi and Taylor, 2018; Woods and Caballero, 2016; Zhang
et al., 2023b). Furthermore, the changes in downward longwave radiation emerge as a crucial factor driving the cold season





Arctic surface warming trend (Zhang et al., 2021b), inspiring us to investigate the specific characteristics of downward longwave radiation closely linked with ARs.

The surface energy budgets of the Arctic represent the net surface heat flux between incoming solar and thermal radiation and
outgoing thermal radiation from the Earth's surface, along with energy exchanges through sensible and latent heat fluxes (Serreze et al., 2007). Short-term perturbations in the surface energy budget, as caused by ARs, may be of climatological significance depending on their magnitude and frequency. These perturbations influence surface warming or cooling, accelerate or decelerate ice growth, and can even contribute to sea ice melting and alter sea ice extent. Therefore, accurate quantification of these energy fluxes is essential for understanding the dynamics of the Arctic climate and its response to
external forcings, such as the influx of moisture-laden ARs.

This study aims to comprehensively quantify the surface energy budget impacts associated with ARs over the Arctic across the entire annual cycle. We seek to unravel the intricate interactions between these atmospheric features and the Arctic surface energy budget by analyzing the spatiotemporal distribution of ARs and associated surface energy budget anomalies.
Furthermore, we intend to quantify the total climatological contributions of ARs to the surface radiative and turbulent heat fluxes and the net surface energy budget of the Arctic when considering the AR occurrence frequency over a 40-year period. Understanding the intricate relationship between ARs and the surface energy budgets provides valuable insights into the mechanisms driving Arctic warming, sea ice melt, and changes in the regional climate.

The study is organized as follows: data and methods are in Section 2. AR's impact on surface energy budgets over the Arctic are analyzed and discussed in Section 3. Sections 4 and 5 provides discussions and conclusions, respectively. This research endeavours to contribute to a deeper understanding of the complex climate dynamics at play in the region through rigorous examination and quantification of the influence of ARs on the surface energy budget in the Arctic.

## 2 Data and Methods

### 2.1 AR detection and tracking

An ensemble Arctic AR index database (Tung et al., 2023) was developed by Zhang et al., (2023a), where a total of 12 AR indices were created based on combinatory conditions of either integrated water vapor transport ($IVT$) or integrated water vapor ($IWV$) applied with three levels of monthly climate thresholds (75th, 85th, and 95th percentiles) using 3-hourly fifth generation of ECMWF atmospheric reanalysis (ERA5, Hersbach et al., 2020) and 3-hourly NASA Modern-Era Retrospective
Analysis for Research and Applications, version 2 (MERRA-2, Gelaro et al., 2017) source data from the AR Tracking Method Intercomparison Project (Shields et al., 2018) from 1980 to 2019. Among these, the $IVT$-based 85th percentile climate threshold index (85th_$IVT$) is consistent with the most commonly adopted indices in AR research (e.g., Guan & Waliser, 2015, 2019;



Ma et al., 2020; P. Zhang et al., 2023). Besides, it has been found that between the two reanalyses (ERA5 and MERRA-2), the 0.25° x 0.25° ERA5 *IVT* field is more precise than the coarser-resolution 0.5° x 0.625° MERRA-2 *IVT* for AR detection and

tracking (Zhang et al., 2023a). Therefore, in this work, we use the 85th_*IVT*-based AR index in ERA5 to analyze the impacts of ARs on the surface energy budgets over the Arctic.

The AR detection and tracking algorithm using the ERA5 85th_*IVT*-based index is briefly summarized as follows, while more detailed descriptions of the ensemble AR detection algorithm can be found in Zhang et al., (2021a, 2023a). First, for each grid

point in the *IVT* field, we select the 3-hourly datum at 1200 UTC each day during neutral or weak El Niño-Southern Oscillation events, grouping the data by month and calculating the 85th percentile for each month across the 40-year period. The resultant monthly 85th percentile values establish a times series of climate thresholds. Next, at each time step, we identify spatial targets as connected grid points with *IVT* values equal or greater than their monthly climate thresholds. We then apply the principal curves method (Hastie and Stuetzle, 1989) to estimate the length of a target formed by aggregating the maximum *IVT* values

at each latitude and longitude within the spatial pattern. The algorithm uses a periodic boundary condition about each latitude. The width of a target is calculated as the total Earth surface area of the identified grid points divided by the length. Any target with length exceeding or equal to 1500 km, while the ratio of length to width greater or equal to 2, is considered as potential presence of an AR object. We then examine the time sequence of the potential AR objects to construct events with the Lagrangian tracking framework based on the spatial proximity and morphological similarity between two consecutive 3-hourly

time steps (Guan and Waliser, 2019). We determine Arctic AR events as those penetrating in the Arctic region (defined as 60°N and northward) and then persisting in the region for at least 18 hours. The AR detection has been facilitated with distributed-parallel computing, specifically, the divide-and-recombine approach using the R-based DeltaRho backended by a Hadoop system (Cleveland and Hafen, 2014; Tung et al., 2018).

**2.2 ERA5 surface fluxes**

In addition to the ERA5 85th_*IVT*-based AR index (Tung et al., 2023; Zhang et al., 2023a), we analyze the surface radiative and turbulent fluxes and near-surface temperature in ERA5. It is noted that ERA5 is the latest reanalysis released by ECMWF in 2019, with high spatial (0.25° longitude by 0.25° latitude) and temporal (hourly) resolutions. It is based on ECMWF Integrated Forecasting System (IFS) CY41r2 by 4D-Var data assimilation and model forecasts with 137 hybrid sigma/model levels to 1 Pa in the vertical (Hersbach et al., 2020). It has been considered a state-of-the-art global reanalysis for the Arctic

(Graham et al., 2019b).

We retrieve hourly surface accumulated fluxes from ERA5 for the 40-year period between January 1980 and December 2019: surface thermal radiation downward (also known as surface downward longwave radiation, LWD), surface net thermal radiation (also known as surface net longwave radiation, LWN), surface net solar radiation (also known as surface net



shortwave radiation, SWN), surface latent heat flux (LH), and surface sensible heat flux (SH). All variables have units of J m$^{-2}$, with the ECMWF sign convention of positive vertical flux values directed downwards towards the surface.

To be consistent with the AR index's 3-hourly instantaneous temporal information and to quantify AR-related surface energy budgets over the Arctic, accumulated surface fluxes (J m$^{-2}$) are converted to hourly mean surface fluxes (W m$^{-2}$), which are
considered as instantaneous surface fluxes at the centre of each hour (i.e., 00:30, 01:30, …, 23:30 UTC). We then further linearly interpolate these values to the start of each hourly period (i.e. 00 UTC, 01 UTC, …, 23 UTC) and downsample the time series to 3-hourly intervals from 00 to 21 UTC to match the temporal resolution of the AR index. Moreover, we define total surface turbulent heat flux (TH) as the sum of SH and LH. The net surface energy budget (SEB) is expressed as the sum of the net radiation at the surface (i.e., sum of the LWN and SWN) and net total TH (i.e., sum of the SH and LH).


In a physical context, a positive net SEB signifies a net transfer of energy from the atmosphere to the surface, while a negative net SEB means the opposite direction of the energy transfer. From a climatological perspective, the net SEB is negative during the Arctic cold season, facilitating surface cooling, sea ice formation, and sensible heat loss from the ocean. In contrast, a positive net SEB dominates the Arctic summer, leading to melting, subsequent reductions in Arctic sea ice, and replenishment
of the ocean's reservoir of sensible heat (Serreze et al., 2007). In addition, we downsample ERA5 instantaneous hourly 2-meter air temperature (T-2m) and surface skin temperature (surface temperature) to the same 3-hourly time intervals from January 1980 to December 2019 to examine AR's warming effects on the surface as done in previous work (e.g., Hegyi and Taylor, 2018; Woods and Caballero, 2016; Zhang et al., 2023b).

## 2.3 Analysis of AR-related surface energy budgets over the Arctic

Composite analyses are performed in order to investigate the impact of the ARs on the surface energy budget components. For each of the surface flux variables, T-2m and surface temperature, calculations are performed at each grid point and for each 3-hourly time step of the day. The 40-year mean climatological values are computed for each day of the year at three hourly frequency and then 3-hourly anomalies for all variables are derived from 1980 to 2019, relative to the corresponding 40-year mean climatology. Given the pronounced seasonality of Arctic ARs outlined by Zhang et al., (2023a), our focus is directed
towards the surface energy budgets associated with ARs during the four distinct seasons: spring (March-May), summer (June-August), fall (September-November), and winter (December-February). Specifically, within each of the four seasons, the 40-year mean climatology values for each 3-hourly time falling within the respective season are obtained at each grid point. Figure 1 illustrates the spatial distribution of AR occurrence frequency in each season. The 40-year mean climatology values of each surface flux variable, T-2m, and surface temperature are visualized in panel (a) of Figs. 2-7. Similarly, the mean anomalies
when AR events occur for each season at each grid point are determined by averaging the anomaly values across all AR occurrences that align with the given season. The corresponding results are displayed in panel (b) of Figs. 2-7 and depict the deviation of the surface fluxes, T-2m and surface temperature from the 40-year climatological mean during AR events.





As discussed earlier, the significant influence of ARs on surface radiative and turbulent fluxes underscores AR's potential role
in driving net SEB fluctuations. Furthermore, these net SEB fluctuations bear a crucial role in governing the extent and volume of sea ice. These considerations motivated our inquiry into the influence of AR occurrences on net SEB anomalies. To quantify the impact of ARs on changing individual components of the SEB and, in turn, the resulting net SEB, we employ a normalization approach that compares the accumulated effect of ARs on surface radiative and turbulent fluxes to the overall climatological SEB. To do this we multiply the mean seasonal AR anomalies for each flux term (panel (b) of Figs. 2-3, 5-7)
by the frequency of AR occurrence (Fig. 1) to account for both the magnitude of the AR anomaly and the occurrence frequency of ARs at any given location. We then normalize this by dividing by the absolute value of the mean SEB (Fig. 7a) to obtain the relative magnitude of the AR impact for each flux term relative to the total SEB. It is important to note that the use of the absolute value of the net SEB is deliberately chosen to maintain consistency with the sign of AR-related SEB term anomalies displayed in panel (b). A positive sign in panel (c) signifies AR's positive contribution to net SEB and therefore warming
effects. The normalization, which accounts for the absolute net SEB, allows for an assessment and comparison of the contributions of AR occurrence in varying the individual SEB components associated with ARs and their subsequent influence on the net SEB.

## 3 Analysis and Results

### 3.1 AR occurrence frequency

Figure 1 visualizes the spatial distributions of 40-year average AR occurrence frequency from 1980 to 2019 during each of the four seasons according to the ERA5 $85^{th}\_IVT$-based index, while Table 1 summarizes the area averaged AR occurrence frequency for four sub-regions – the central Arctic (including the Barents and Kara Seas), sub-polar oceans, continents, and Greenland (Fig. S1). The distribution of AR occurrences exhibits prominent seasonality and regional characteristics. The central Arctic Ocean has the lowest AR frequency in the study domain ranging from 10.4% (summer) to 10.8% (spring). AR
frequency in the other sub-regions ranges from 11.1 to 12.7% with the seasonal lowest values (11.1 to 11.8%) in summer and frequencies usually above 12% in the fall, winter and spring. The seasonal change in AR frequency in sub-polar latitudes parallels changes in storm track intensity while the lower values in the central Arctic reflect the lower frequency of storms at high latitudes year-round (Valkonen et al., 2021). As seen in Fig.1, over sub-polar regions, the maximum AR occurrence frequency shifts from the Arctic Atlantic sector stretching from the Barents-Kara Sea to central Siberia in fall and winter to
the Greenland-Labrador Sea and the Arctic Pacific sector (i.e., Bering Sea) extending towards Alaska and western Canada in spring, with the greatest seasonal magnitudes in summer concentrated over the Labrador Sea. Around the Arctic pole, ARs are least in winter, gradually increasing from spring to fall, which possibly reflects the seasonal poleward shift of jet streams in summer but retreat equatorward in winter, along with colder polar temperature reducing water vapor contents and influencing AR occurrence in winter.



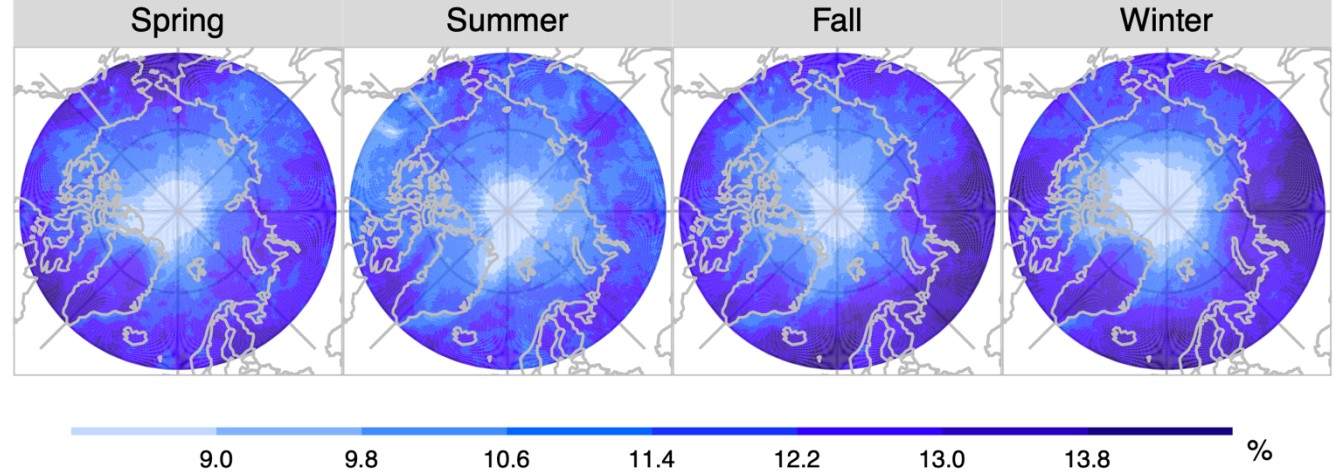


**Figure 1: Spatial distributions of 40-year average AR occurrence frequency (percentage of AR occurrence time steps, unit: %) at each of the four seasons: Spring (March-May), Summer (June-August), Fall (September-November) and Winter (December-February), from January 1980 to December 2019.**

## 3.2 AR's influence on the surface energy budget components of the Arctic

Our initial investigation focuses on assessing the impacts of ARs on the surface radiation components and the corresponding surface responses, including surface downward longwave radiation (LWD, Fig. 2), net surface longwave radiation (LWN, Fig. 3), T-2m and surface temperature (Fig. 4), and net surface shortwave radiation (SWN, Fig. 5). We subsequently analyze the influences of ARs on surface turbulent flux (TH, Fig.6) and the net surface energy budget (SEB, Fig. 7). As detailed in section 2.3, for each of the analyzed SEB terms, our analysis comprises three aspects: calculating the average climatology during each

season (panel (a)), determining composite anomalies during the presence of ARs (panel (b)), and establishing the total contributions from each radiation term to absolute net SEB due to the occurrence of ARs (panel (c)). The regional average results for each season can be found in Table 1 and Table S1, which also indicates how the relative AR contribution to the net SEB compares to the AR frequency.

### 3.2.1 Surface radiative flexes

The 40-year climatological LWD in the central Arctic (Fig. 2a and Table 1) exhibits strong seasonality with the highest values in summer (297 W m$^{-2}$) and the lowest in winter (188 W m$^{-2}$). There is also a clear meridional gradient with larger values at lower latitudes, except Greenland, than at higher latitudes. Additional spatial variability comes from land – sea contrasts with larger values over open water in the cool seasons compared to over land or sea ice. On average, LWD over sub-polar oceans is more than 40 W m$^{-2}$ greater than that over land and sea ice in winter (Table 1). Finally, high elevation locations, such as the

Greenland ice sheet, have less LWD due to the colder and drier atmosphere at the surface over high terrain.







**Figure 2. Maps showing (a) the spatial distributions of 40-year mean surface downward longwave radiation (LWD, unit: W m⁻²)**
**across Spring (March-May), Summer (June-August), Fall (September-November), and Winter (December-February) from 1980 to**





**2019. (b) spatial distributions of 40-year mean LWD anomalies (unit: W m⁻²) during the presence of AR events within each season.**

**(c) Spatial distributions of the fraction of 40-year AR contribution (unit: %) from the total LWD anomalies to the absolute values of 40-year mean SEB for each season. The percentage results greater than 100% are shaded in grey for clarity. Note that all positive values of fluxes are directed downwards at the surface.**

| Surface Downward Longwave Radiation (LWD) | | | | | | | | |
|---|---|---|---|---|---|---|---|---|
| Region | Central Arctic | Subpolar Oceans | Continents | Greenland | Central Arctic | Subpolar Oceans | Continents | Greenland |
| Season | Spring | | | | Summer | | | |
| AR Freq. (%) | 10.8 | 12.3 | 12.2 | 12.2 | 10.4 | 11.8 | 11.6 | 11.1 |
| Climatology (W m⁻²) | 220.6 | 250.5 | 229.1 | 177.2 | 297.2 | 306.5 | 315.1 | 234.8 |
| Anomalies (W m⁻²) | 32.9 | 36.2 | 38.2 | 43.8 | 14.6 | 21.4 | 23.6 | 34 |
| Cotrib. to SEB (%) | 45.1 | 68.7 | 213.5 | 4759.7 | 2.3 | 2.6 | 18.8 | 165.2 |
| Extra AR (%) | 34.3 | 56.4 | 201.3 | 4747.5 | -8.1 | -9.2 | 7.2 | 154.1 |
| Season | Fall | | | | Winter | | | |
| AR Freq. (%) | 10.6 | 12.4 | 12.5 | 11.8 | 10.5 | 12.3 | 12.7 | 12.4 |
| Climatology (W m⁻²) | 246.5 | 274.3 | 249.4 | 190.2 | 187.6 | 230.9 | 188.1 | 159.9 |
| Anomalies (W m⁻²) | 29 | 34.9 | 36.5 | 46.6 | 45.1 | 43.8 | 45.9 | 47.3 |
| Cotrib. to SEB (%) | 8.3 | 5.9 | 56.8 | 100.9 | 9.7 | 5.6 | 168.9 | 133.6 |
| Extra AR (%) | -2.3 | -6.5 | 44.3 | 89.1 | -0.8 | -6.7 | 156.2 | 121.2 |

| Net Surface Energy Budget (SEB) | | | | | | | | |
|---|---|---|---|---|---|---|---|---|
| Region | Central Arctic | Subpolar Oceans | Continents | Greenland | Central Arctic | Subpolar Oceans | Continents | Greenland |
| Season | Spring | | | | Summer | | | |
| AR Freq. (%) | 10.8 | 12.3 | 12.2 | 12.2 | 10.4 | 11.8 | 11.6 | 11.1 |
| Climatology (W m⁻²) | -19.6 | -21.7 | 9.3 | 0.4 | 69.8 | 103.8 | 21.9 | 12.8 |
| Anomalies (W m⁻²) | 25.8 | 39.8 | 15.3 | 20 | 8.9 | -8 | 3.2 | 10.3 |
| Cotrib. to SEB (%) | 31.8 | 65.3 | 89.5 | 2199.2 | 1.2 | -0.8 | 2.7 | 62.5 |
| Extra AR (%) | 21 | 53 | 77.3 | 2187 | -9.2 | -11 | -8.9 | 51.4 |
| Season | Fall | | | | Winter | | | |
| AR Freq. (%) | 10.6 | 12.4 | 12.5 | 11.8 | 10.5 | 12.3 | 12.7 | 12.4 |
| Climatology (W m⁻²) | -48 | -89.8 | -13 | -11.5 | -64.6 | -146.9 | -10.1 | -12.9 |
| Anomalies (W m⁻²) | 34.4 | 64.1 | 16 | 24.9 | 39.2 | 90.9 | 15.6 | 27.7 |
| Cotrib. to SEB (%) | 7.9 | 9.4 | 23.6 | 54.4 | 6.9 | 7.9 | 49.9 | 79.5 |
| Extra AR (%) | -2.7 | -3 | 11.1 | 42.6 | -3.6 | -4.4 | 37.2 | 67.1 |

**Table 1. Regional average results of Surface Downward Longwave Radiation (top panel) and Net Surface Energy Budget (SEB,**

**bottom panel) across different seasons: Spring (shaded in green), Summer (shaded in orange), Fall (shaded in yellow), and Winter (shaded in blue). Results include AR occurrence frequency (AR Freq., unit: %), Climatology (unit: W m-2), composite anomalies**



**(Anomalies, unit: W m-2), total AR contribution to absolute net SEB (Contrib. to SEB, unit: %), and relative AR contribution to the net SEB compares to the AR frequency (Extra AR, unit: %).**

Across the study domain ARs produce discernibly positive LWD anomalies, likely attributed to enhanced water vapor content associated with ARs and cloud formation (particularly low-level liquid clouds, Shupe and Intrieri, 2004). Unlike the seasonality in Fig. 2a, AR-related surface LWD anomalies (Fig. 2b) are largest in winter and similar across all sub-regions, with area-averaged anomalies in excess of 40 W m$^{-2}$ (Table 1). The AR-related surface LWD anomalies are smallest in summer, ranging from 15 W m$^{-2}$ over the central Arctic to 34 W m$^{-2}$ over Greenland, with sub-polar ocean and land areas having similar LWD

anomalies (21 to 24 W m$^{-2}$). Spring and fall have AR LWD anomalies that are slightly smaller than those seen in winter, with Greenland having the largest anomalies (44 to 47 W m$^{-2}$) followed by sub-polar ocean and land areas (35 to 38 W m$^{-2}$) and the central Arctic (29 to 33 W m$^{-2}$). Overall, the large LWD anomalies in winter and small anomalies in summer, is potentially related to the previous finding that the highest sensitivity of longwave cloud forcing to liquid water path corresponds to clouds with low liquid water path, and sensitivity decreases as liquid water path increases (Chen et al., 2006). In winter, Arctic clouds

have relatively low amounts of liquid water, so an increase in cloud liquid water associated with Arctic ARs leads to a larger LWD. In addition, the winter AR LWD may be more connected to clear-sky LWD than that in summer, given the dry and cold Arctic conditions. A previous study suggests that clear-sky LWD plays a more prominent role in contributing to surface warming during cold seasons when conducting an Arctic SEB analysis (Zhang et al., 2021b).

ARs consistently induce some of the largest positive LWD anomalies of anywhere in the study domain over western and southern Greenland throughout the year (Fig. 2b). The location of the largest LWD anomalies over portions of the Greenland ice sheet is consistent with previous work that has suggested ARs play an important role in triggering melt events over the ice sheet (Mattingly et al., 2018, 2023, 2020; Neff, 2018; Neff et al., 2014). Moreover, during the cold seasons (fall, winter, and spring), the largest LWD anomalies over continents are observed east of the Ural Mountains, western Alaska, eastern Siberia,

and portions of the Canadian Archipelago. Over sub-polar oceans, the largest LWD anomalies in cold seasons occur in the Greenland-Barents Seas and Bering-Chukchi-Beaufort Seas, extending between these two maxima particularly in winter and, to a lesser extent, in spring. These large LWD anomalies, particularly near the sea ice edge in the northern Greenland and Barents Seas, the Kara Sea and in the Chukchi and Beaufort Seas, is likely an important driver for on-going changes in sea ice extent and volume. Specifically, the large LWD anomalies may serve to initiate sea ice melt in spring, delay the onset of ice

formation in the fall, and slow ice growth in the winter (Huang et al., 2019b, a; Park et al., 2015a; Zhang et al., 2023b). These impacts in sea ice extent and volume likely contribute to increased water vapor and the emergence of associated LWD anomalies, as observed in Fig. 2b.

The pronounced impact of ARs on surface LWD motivated us to explore their contribution to the mean SEB. The contributions

of seasonally integrated AR LWD anomalies to the mean SEB are depicted in Fig. 2c. Large contributions seen in Fig. 2c are



due to a combination of large AR LWD anomalies (Fig. 2b) and high AR frequency (Fig. 1) but can also happen when the mean SEB is small (Fig. 7a). The locations of small SEB often have AR contributions to the SEB that exceed 100% and these locations are shaded in grey for clarity in Fig. 2c and include regions such as Greenland and central Eurasia in spring and winter.


AR associated LWD anomalies make positive contributions to the net SEB (Fig. 2c) although their relative contribution to the net SEB does not always exceed their frequency of occurrence (Table 1, last row showing extra AR contribution). The AR LWD anomalies make the most substantial contribution to the net SEB over the North American and Eurasian continents and Greenland across all seasons (19 to >200% of the net SEB), with the relative contribution of ARs far exceeding their frequency of occurrence suggesting that ARs play an outsized role in the net SEB in these regions relative to how often they occur. This pattern is attributed to the large (Greenland) to moderate (continents) AR LWD anomalies and frequencies combined with comparatively smaller net SEB prevalent over land areas (Fig. 7a). Conversely, there are small relative contributions of AR-related LWD to the net SEB over sub-polar open water regions in fall and winter (<6%), which is less than the AR frequency of occurrence, owing to the large net SEB associated with open water at these times of the year (Fig. 7a). AR-related LWD contribution to net SEB over sub-polar oceans is largest in spring (69%) due to the reduced net SEB over open water in this season, while summer has the smallest AR-related LWD contribution to the net SEB (3%), as a result of the smaller AR LWD anomalies in this season. Over the central Arctic Ocean, different from the seasonality observed in the AR-related LWD anomalies shown in Fig. 2b, AR's total contribution of LWD anomalies to the net SEB are largest in spring (45%), far exceeding the AR frequency of occurrence. In contrast, AR contribution to the net SEB is small (<10%) during the rest of the year and is less than the corresponding AR frequency.

Prior research has demonstrated the crucial role of springtime clouds and their associated cloud-radiative effects in dictating the onset of sea ice melt and the eventual extent of fall sea ice (Huang et al., 2019b). Our findings reinforce these observations, indicating that the presence of ARs and associated augmented clouds, and the accompanying enhanced LWD, likely play an important role in the initiation of sea ice melt in spring via the large AR contribution to the net SEB. Considering the opposite signs of surface energy budgets between the colder seasons (fall, winter, and spring) and summer (Fig. 7a), AR-related LWD expedites the sea ice melt in summer, although Fig. 2c suggests that this may be a relatively minimal impact. Conversely, in fall and winter, AR's positive contributions contradict the negative net SEB (Fig. 7a), implying the potential for delaying the sea ice recovery.

285

Over the central Arctic, the analysis of AR-related LWD anomalies presented above (Fig. 2) highlights the large absolute impact of ARs during the winter, with anomalies in excess of 40 W m$^{-2}$ (Table 1). However, ARs make their most significant relative contribution to the average net SEB in spring, accounting for at least 45% of the net SEB, surpassing the corresponding AR frequency by more than 34% (Table 1). While ARs and analogous poleward moisture intrusions have traditionally received





290 considerable attention during the winter period, our study highlights a distinctive finding. When normalized by the net SEB the most substantial contributions from AR-related LWD anomalies to the net SEB are realized during the spring season, when the melt of sea ice is initiated.

 The climatological net surface longwave radiation (LWN) in the Arctic (Fig. 3a and Table S1) is consistently negative, cooling

295 the surface and exhibiting distinct seasonal variations and spatial land–sea-ice contrast, influenced by surface temperature (Fig. 4a) and LWD (Fig. 2a). In the sub-polar regions, the maximum negative LWN shifts from the land in summer (-53 W m$^{-2}$ for continents and -51 W m$^{-2}$ for Greenland) to open water in fall (-51 W m$^{-2}$) and winter (-56 W m$^{-2}$), with similar values over sub-polar land and water in spring (< -47 W m$^{-2}$). The central Arctic Ocean displays a unique seasonality: the least LWN cooling occurs in summer (-23 W m$^{-2}$), reaching its peak in winter (-44 W m$^{-2}$), while spring (-40 W m$^{-2}$) and fall (-34 W m$^{-2}$)

300 have intermediate values.

 The large positive LWD anomalies associated with ARs (Fig. 2b) are partially counteracted by the opposing surface upward longwave radiation (LWU) anomalies, resulting in consistent positive but smaller magnitudes of LWN anomalies across the Arctic study domain (Fig. 3b) ranging from 12 to 31 W m$^{-2}$ (Table S1). The distribution of AR LWN anomalies exhibits a

305 land-sea contrast in each season. In winter, the smaller response of sub-polar oceanic SSTs to the presence of ARs, compared to land or sea ice regions (Fig. 4b), results in small LWU anomalies, and therefore amplified LWN anomalies, with a regional average of 31 W m$^{-2}$. The next largest LWN anomalies in winter are over the central Arctic Ocean (22 W m$^{-2}$) and are driven by large LWD anomalies (Fig. 2b) offset by moderate increases in surface temperature (Fig. 4b). In winter, AR LWN anomalies are smallest over Greenland (18 W m$^{-2}$) and land areas (12 W m$^{-2}$) due to the large AR-associated increase in surface

310 temperature in these regions (Fig. 4b). In summer, the land-sea contrast of AR LWN anomalies diminishes, decreasing to 21 W m$^{-2}$ for sub-polar oceans and 14 W m$^{-2}$ for the central Arctic Ocean but increasing to 17 W m$^{-2}$ and 22 W m$^{-2}$ for continents and Greenland, respectively. Spring and fall have a similar spatial distribution of AR LWN anomalies, but smaller magnitudes compared to winter over the sea and summer over the land.



**Figure 3, similar to Figure 2, but for the results according to the net surface longwave radiation (LWN).**





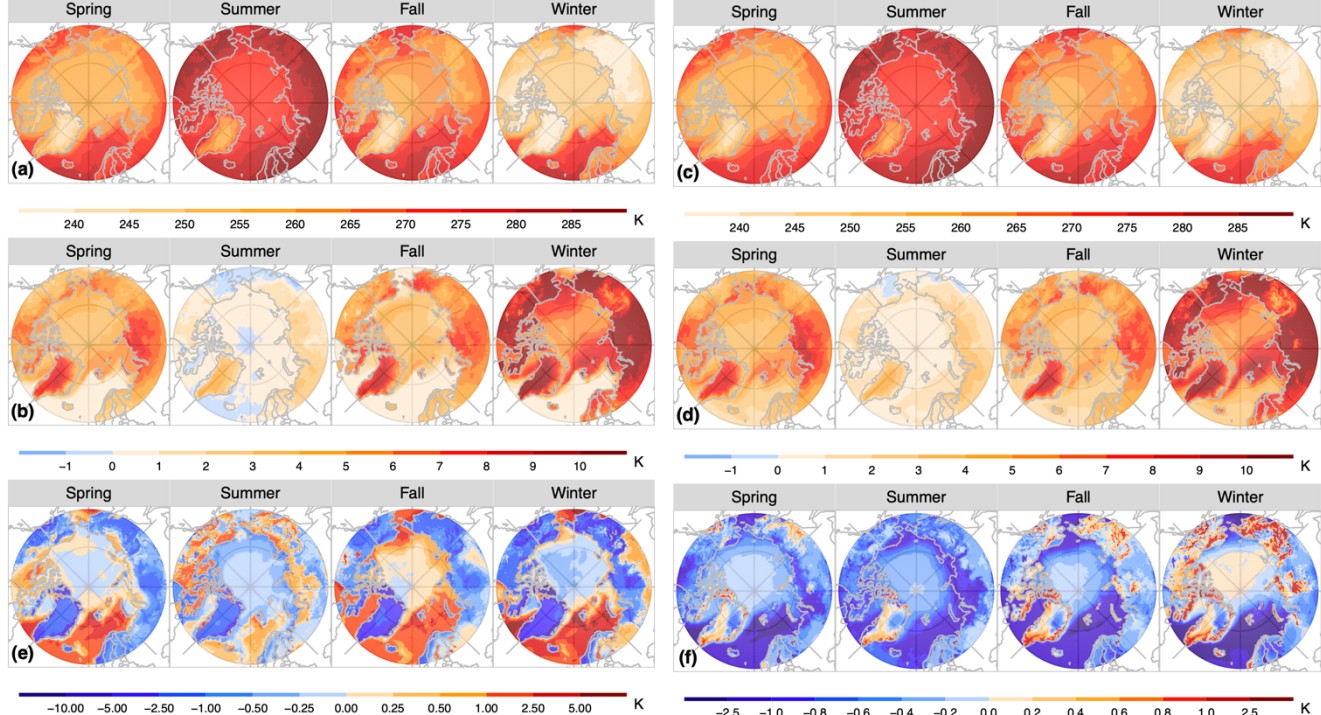

**Figure 4. (a) spatial distributions of 40-year mean surface skin temperature (surface temperature, unit: K) during each of the four seasons: Spring (March-May), Summer (June-August), Fall (September-November) and Winter (December-February) from 1980 to 2019. (b) spatial distributions of 40-year mean surface skin temperature anomalies (unit: K) during the presence of AR events during each season from 1980 to 2019. (c), (d) are similar to (a), (b), but for results of T-2m. (e) The differences (unit: K) between the 40-year mean surface temperature in (a) and the 40-year mean T-2m in (c), shown as $surface\ temperature - T2m$. (f) The differences (unit: K) between the 40-year mean surface temperature anomalies associated with ARs in (b) and the 40-year mean T-2m anomalies associated with ARs in (d), shown as $anomalies\ in\ surface\ temperature - anomalies\ in\ T2m$.**

As seen in Fig. 3b, over the central Arctic Ocean, the impacts of ARs on LWN are particularly conspicuous within the marginal ice zone, such as the Barents-Kara seas in winter and to a lesser extent in fall, and the pack ice area in spring. The local maximum AR-related positive LWN anomalies signify that the presence of Arctic ARs potentially trigger melting and diminishing sea ice coverage within these marginal sea ice regions (Boisvert et al., 2016) and retarding the thickening process of the ice layer over pack ice areas (Persson et al., 2017), particularly during winter and spring when the sea ice is maximum. The impacts of ARs on the LWN are smaller in summer. Consistent with AR LWD anomalies observed in Fig. 2b, ARs generate large LWN anomalies over western and southern Greenland, supporting their crucial role in triggering melt events over the Greenland ice sheet (Mattingly et al., 2018, 2020).



When normalized by the mean net SEB (Fig. 7a), the spatial distribution of total anomalies in LWN associated with ARs, as
shown in Fig. 3c, resembles the pattern in Fig. 2c, but with reduced magnitudes (Table S1). Still, the most significant AR
contributions are observed over regions with small mean SEB (Fig. 7a), such as Eurasian and North American continents and
Greenland, throughout the year (13 to >100% of the net SEB). The large relative contribution, far exceeding their
corresponding AR occurrence frequency, further implies that ARs play a crucial role in the net SEB in these regions,
particularly in spring. Although AR LWN anomalies have larger absolute magnitudes over sub-polar oceans (Fig. 3b), their
relative contribution to net SEB significantly decreases year-round due to the larger mean SEB (Fig. 7a). AR-related LWN
contribution to net SEB over sub-polar oceans is most pronounced in spring (49%), primarily due to the reduced net SEB over
open water in this season, surpassing the corresponding AR frequency by 37%. In contrast, summer and winter exhibit the
smallest AR-related LWD contribution to the net SEB (3%) due to smaller AR LWN anomalies in summer and a larger mean
net SEB in winter, both falling below the corresponding AR frequency by 9%. As the mean net SEB decrease in fall, the
relative contribution slightly increases to 5%, still below the corresponding AR frequency. Over the central Arctic Ocean,
AR's total contribution of LWN anomalies to the net SEB is highest in spring (21%), exceeding the AR occurrence frequency.
However, it decreases to 2-4% in other seasons, with this percentage falling below the corresponding AR frequency.

The discussion above highlights the interplay between AR-induced LWD anomalies and the surface temperature response to
ARs in determining AR impact on net LW. The small response of sub-polar ocean SST to ARs results in ARs having the
largest (or near largest in summer) absolute impact on net LW anomalies in this region throughout the year. However, because
the net SEB is large over the sub-polar oceans, the AR induced net LW contribution to the net SEB is relatively small, and less
than the AR frequency, except in spring. Over continental land areas AR induced net LW anomalies are smaller than over sub-
polar oceans or the central Arctic, except in summer when the anomalies are larger over land than the central Arctic, due to
the large AR-induced warming of surface temperature (Fig. 4b). Because the net SEB is small over land, the AR net LW
contribution to the net SEB is large and consistently exceeds the AR frequency, suggesting that ARs play an important role in
shaping the net SEB over land areas. Similar results, with larger AR-induced anomalies, are seen over Greenland and
emphasize that ARs are an important factor in melt events over Greenland. The results over the central Arctic are more
complex. AR-induced net LW anomalies are intermediate to those over sub-polar oceans and land in fall, winter and spring,
ranging from 16 to 22 W m$^{-2}$. This is because AR-induced LWD anomalies are large (Fig. 2b) but so is the surface temperature
response to ARs (Fig. 4b). The AR-induced net LW contribution to the net SEB is small (~4%) in fall and winter and less than
the AR frequency. This indicates that in terms of net LW, ARs are not important to the net SEB in the central Arctic, despite
their large contribution to LWD, due to their significant surface warming signal. In contrast, in spring AR-induced net LW
anomalies contribute 21% to the net SEB, far exceeding AR frequency at this time of year, suggesting that ARs are important
leading into the sea ice melt season and may serve to initiate melt. In summer, small AR-induced LWD anomalies and surface
warming result in the smallest seasonal net LW anomalies in the central Arctic (14 W m$^{-2}$) with a minimal contribution to the



net SEB (2%) that is much smaller than the AR frequency, suggesting that ARs are not important in terms of net LW in the central Arctic at this time of year.

The net surface shortwave radiation (SWN) received at the Arctic surface (Fig. 5a and Table S1) displays significant seasonal variation, with the highest values occurring during summer (>65 W m$^{-2}$) and the lowest during winter (<6 W m$^{-2}$), primarily attributable to the limited daylight hours and low solar angles in winter. A noticeable meridional gradient is observed, with the maximum SWN in sub-polar continental and ocean regions in summer (>150 W m$^{-2}$), due to higher sun angle and lower albedo. SWN is less over Greenland (65 W m$^{-2}$) and in the central Arctic (100 W m$^{-2}$) in the summer due to the higher surface albedo.

During the transition seasons, the Arctic receives a greater overall SWN in spring as compared to the fall.

Across the entire Arctic, ARs exhibit negative SWN anomalies, as depicted in Fig. 5b, primarily attributed to the enhanced cloud formation, mainly low-level thick clouds, which reflect solar radiation and reduce SEB. The AR-induced decreasing SWN is most pronounced in summer when the solar radiation is at the maximum (Fig. 5a). A meridional gradient is observed,

with larger anomaly magnitudes in lower albedo sub-polar regions and lower magnitudes in areas of sea ice-covered Arctic Ocean and the Greenland Ice Sheet, which have high albedo, resulting in attenuated SWN effects associated with ARs. In summer, the regional average anomaly ranges from -17 W m$^{-2}$ over Greenland and -22 W m$^{-2}$ over the Central Arctic Ocean to -35 and -52 W m$^{-2}$ for continents and sub-polar oceans, respectively. AR-related SWN effects are attenuated in the sea ice-covered Arctic Ocean and the Greenland Ice Sheet, possibly due to the similarity between the albedo of the sea ice-covered

surface and the clouds associated with ARs, resulting in comparable SWN (Curry and Ebert, 1992; Miller et al., 2015). The spatial variations from land-sea-ice contrasts noted in summer are also evident in spring, but to a lesser extent, with anomalies ranging from -6 to -29 W m$^{-2}$. Due to limited SWN in fall and winter, AR-related SW impacts are significantly reduced.

The SWN from ARs make negative contributions to the net SEB, primarily during spring, somewhat less in summer and

limited in fall and winter (Fig. 5c). Still, due to the smaller mean net SEB (Fig. 7a), the most pronounced negative contribution occurs over the continents and Greenland in spring (-67 and - 554%) and summer (-27 and -47%) with the magnitude of these contributions greatly exceeding the AR frequency. AR SWN contributions to the net SEB are much smaller in fall (-12 and - 5 %) and winter (-8% and -1%), with percentage values lower than their corresponding AR frequency during these seasons. In the spring over sub-polar oceans, the substantial AR-related SWN anomalies (Fig.5b), combined with a higher AR occurrence

frequency (Fig. 1), result in a larger contribution to the mean SEB, averaging -51% and exceeding the corresponding AR frequency by 38%. This average decreases dramatically to -6% in summer, and below -2% in fall and winter, all falling below the corresponding AR frequency. Over the central Arctic Ocean, AR SWN anomalies contribute -8% in spring and -4% in summer to the net SEB, although both numbers are lower than the corresponding AR frequency. Fig. 5c reveals local maximum contributions ranging from 4% to 8% over the central Arctic Ocean in summer, emphasizing AR's cooling impact on SWN

and their role in slowing sea ice melt. AR-induced SWN contributions are negligible (<1%) during fall and winter.





**Figure 5, similar to Figure 2, but for results according to the net shortwave radiation (SWN).**



Comparing AR-related LWN and SWN anomalies (Figs. 3 and 5, Table S1) shows that in summer AR-related SWN cooling effects dominate surface radiation budgets over the low albedo sub-polar oceans and continents. Over the higher albedo Greenland ice sheet and central Arctic, AR-related LWN and SWN anomalies differ by less than 1 W m$^{-2}$, indicating little overall radiative impact of ARs at this time of year over ice-covered surfaces. In contrast, ARs play a warming role during the cold seasons (spring, fall and winter) over Greenland and the central Arctic with AR LWN exceeding the corresponding AR

SWN in both absolute anomalies and relative contribution. ARs also result in radiative warming over the sub-polar oceans and continents in fall and winter with a slight cooling effect in spring. Notably, AR-related LW radiative warming and SW radiative cooling effects in spring are crucial to influence the fall sea ice extent, depending on their magnitudes (Cox et al., 2016).

### 3.2.2 Surface turbulent heat fluxes

Turbulent heat (TH) fluxes are a crucial component of the surface energy balance in the Arctic and plays a significant role in

influencing the regional climate, sea ice changes and atmospheric circulation patterns (Bourassa et al., 2013). Turbulent sensible (SH) and latent heat (LH) fluxes constitute essential components of the energy exchange between the Earth's surface and the overlying atmosphere through turbulent mixing and eddy processes.

The 40-year climatological Arctic TH (Fig. 6a and Table S1) exhibits significant regional seasonality characterized by contrasts

between land, sea, and sea ice. In summer, the relatively warmer and wetter Arctic surface drives upward SH and LH, resulting in upward (negative) TH across most of the Arctic (excluding Greenland). The regional averages range from -81 W m$^{-2}$ over continents, -16 W m$^{-2}$ over sub-polar oceans, to -7 W m$^{-2}$ over the central Arctic. In winter, the radiatively cooled surface results in near-surface temperature inversions (Fig. 4e) over much of the land, including the Greenland ice sheet. As a result, the continents and Greenland experience downward TH (11 and 24 W m$^{-2}$ respectively), from stronger downward SH (Fig.

S2a) and weak downward LH (Fig. S3a). In contrast, the relatively warmer sub-polar oceans exhibit stronger upward TH, averaging -96 W m$^{-2}$, with the most intense areas extending to the Barents-Kara Sea. The sea ice covered portions of the central Arctic Ocean (Fig. 6a), like land areas and Greenland, experiences downward TH in winter. This is due to the presence of a near-surface temperature inversion (Fig 4e), resulting in a downward SH (Fig. S2a), offset slightly by a weak upward LH (Fig. S3a) due to limited surface evaporation with lower moisture content. Despite the downward TH over the ice-covered portions

of the Arctic Ocean the area averaged TH for the central Arctic region, which includes the Barents and Kara Seas, is upward (-21 W m$^{-2}$) due to the large upward fluxes in these seas. The two transition seasons, spring and fall, display patterns more similar to summer but with reduced magnitudes over continents and an increase in the central Arctic and sub-polar oceans. In fall, winter, and spring, the cold and dry Greenland ice sheet consistently experiences downward TH (13 to 24 W m$^{-2}$), with a small upward heat flux (-1 W m$^{-2}$) in summer.





**Figure 6, similar to Figure 2, but for results according to the turbulent heat flux (TH).**





In Fig. 6b and Table S1, ARs induce pronounced positive (downward) TH anomalies across most of the Arctic region, primarily attributed to the increased temperature and moisture associated with ARs overlying the cold Arctic surface. Consistent with Fig. 6a, the AR-related anomalies exhibit seasonal variability and distinct land – sea – sea ice contrasts. The most substantial impact of AR TH is evident over sub-polar oceans, particularly during the colder seasons. Unlike other Arctic regions the surface temperature over the sub-polar oceans exhibits only very weak warming during AR events (Fig. 4b) but the presence of near surface warm air associated with the ARs (Fig. 4d) weakens the climatological ocean-to-air temperature gradient (warmer ocean and colder air) (Figs. 4e and 4f), causing a reduction (42 to 62 W m$^{-2}$) in the normally large upward TH (-58 to -96 W m$^{-2}$). In winter, other regions display much weaker positive (downward) TH anomalies (18 W m$^{-2}$ over central Arctic, 10 W m$^{-2}$ over Greenland, 6 W m$^{-2}$ over continents). Unlike over the sub-polar oceans, the surface temperature over land and sea ice-covered regions exhibits strong warming during AR events. This surface warming, in conjunction with similar magnitude near-surface warming (Fig. 4d), results in areas of both strengthened and weakened near-surface temperature gradients (Fig. 4f) that differ from the pronounced weakening of the near-surface temperature gradient over the sub-polar oceans. As a result, the AR-related TH anomalies are much smaller over the land and sea ice covered regions than over areas of open water in the winter. Similar patterns are seen in fall and spring with reduced magnitude TH anomalies over Greenland (7 W m$^{-2}$), similar magnitude TH anomalies over the central Arctic Ocean (15-21 W m$^{-2}$), and larger anomalies over continents (15-10 W m$^{-2}$) compared to the winter anomalies. In summer, the AR-related downward TH anomalies are much weaker over the sub-polar oceans (23 W m$^{-2}$) compared to the colder seasons, but still larger than land or sea ice regions. The continental areas have the next largest downward TH anomalies in summer (21 W m$^{-2}$) driven by stronger warming of the near surface air (Fig. 4d) relative to the ground (Figs. 4b and 4f). Over the sea ice-covered portion of the Arctic Ocean the surface temperature exhibits little warming during AR events (Fig. 4b), since the ice surface is already near the melting point, while the near-surface air warms (Figs. 4d and 4f), resulting in a moderate downward TH anomaly (17 W m$^{-2}$). The Greenland ice sheet experiences the smallest AR-related TH anomaly in summer (6 W m$^{-2}$).

Figure 6c and Table S1 reveals that, as anticipated, AR TH anomalies make large relative contributions to the mean SEB over continents and Greenland, with the maximum in spring (61% and >100%) and similar percentages across the other seasons (16-17% and 7-28%). In all seasons the AR TH anomaly contribution to the mean SEB exceeds their corresponding AR occurrence frequency. While the largest AR TH anomalies are seen over the sub-polar oceans (Fig. 6b), the contribution of these anomalies to the mean SEB is much smaller (Fig 6c), due to the large climatological net SEB over the oceans (Fig. 7a). In summer, fall and winter the AR TH anomalies contribute 3% to 7% to the mean SEB and these contributions are less than the corresponding AR frequency. The largest contribution of AR TH anomalies to the mean SEB over the sub-polar oceans is 67% in spring, far exceeding the corresponding AR frequency by 55%. Over the central Arctic Ocean, AR's total contribution of TH anomalies to the mean SEB is largest in spring (19%), exceeding the AR frequency by 8%, but small during the rest of the year (3-4%) and less than the AR frequency. Additionally, localized maximum relative contribution are observed along the





sea ice margins (e.g., Chukchi Sea, Barents-Kara-Laptev Seas), with the maximum in spring. This is primarily attributed to the higher AR occurrence frequency (Fig. 1) and the local maximum in TH anomalies (Fig. 6b).

The analysis of AR-related TH anomalies in Fig. 6 highlights AR's substantial absolute impact over the sub-polar oceans, particularly in the cold seasons, which exceeds the corresponding LWD anomalies in Fig. 2b. Notably, their relative contribution to the net SEB is most pronounced in spring, far exceeding the corresponding AR frequency. Moderate AR-related TH anomalies are seen year-round over continental areas and the Greenland ice sheet (Fig. 6b). The contribution of these AR-related TH anomalies to the mean SEB are large in these regions, exceeding the AR frequency. In the central Arctic Ocean, there are moderate AR-related TH anomalies (Fig. 6b), but their contribution to the mean SEB exceeds the AR frequency only

in the spring (Table S1). However, there are local maxima in both absolute AR-related TH anomalies (particularly in summer) and their corresponding relative contribution (particularly in spring), located over the sea ice margins extending to the sub-polar oceans.

### 3.2.3 Net surface energy budget

The pronounced seasonality of the Arctic SEB components (Fig. 2a, 3a, 5a, 6a), characterized by spatial land - sea - sea ice

contrast, leads to a distinctive regional seasonality of net SEB, as shown in Fig. 7a. In summer, dominated by a substantial SWN, positive net SEB is observed across most of the study domain, ranging from 104 W m$^{-2}$ over sub-polar oceans, 70 W m$^{-2}$ over central Arctic, to 22 W m$^{-2}$ and 13 W m$^{-2}$ over continents and Greenland, respectively (Table 1). In contrast, in winter (and to a lesser extent in fall), when SW is at a minimum, the net SEB is primarily driven by LWN and TH. This results in a net upward SEB flux with average values ranging from -147 W m$^{-2}$ (-90 W m$^{-2}$) over sub-polar oceans, -65 W m$^{-2}$ (-48 W m$^{-2}$)

over central Arctic, to -10 W m$^{-2}$ (-13 W m$^{-2}$) and -13 W m$^{-2}$ (-12 W m$^{-2}$) over continents and Greenland, respectively. Spring displays a distinctive spatial variability, with weak downward net SEB flux over continents (9 W m$^{-2}$) and near-zero fluxes over Greenland but surface energy loss (upward net SEB) over the sub-polar oceans (-22 W m$^{-2}$) and central Arctic (-20 W m$^{-2}$). Additionally, distinct variations in the sea ice margins of the central Arctic Ocean (Fig. 7a) are observed year-round, with larger positive net SEB values contributing to summer sea ice melting and larger negative net SEB values leading to refreezing

from fall, winter, to spring.



**Figure 7, similar to Figure 2, but for the results according to the net surface energy budgets (SEB).**





ARs induce discernible net SEB anomalies across the entire Arctic domain, predominantly in cold seasons, with a distinct yet
weak impact observed in summer (Fig. 7b). The spatial distributions of AR-induced net SEB anomalies exhibit pronounced
land - sea - sea ice contrasts. In cold seasons, the most pronounced positive impacts are observed over sub-polar oceans,
ranging from 91 W m$^{-2}$ in winter to 64 W m$^{-2}$ and 40 W m$^{-2}$ in fall and spring (Table 1). These positive anomalies are primarily
driven by substantial positive TH anomalies (Fig. 6b) and positive LWN anomalies (Fig. 3b), greatly surpassing weaker
negative SW impacts (Fig. 5b). The intensified positive net SEB anomalies extend northward to the sea ice margins of the
Arctic Ocean. Much smaller AR-related net SEB anomalies are seen over the central Arctic, averaging between 26 W m$^{-2}$ and
39 W m$^{-2}$. This potentially hinders sea ice refreezing in fall and winter and may trigger sea ice melt in spring. AR-induced net
SEB anomalies are notably weaker over the continents and Greenland (15-28 W m$^{-2}$) and are dominated by the LWN AR
anomalies. In summer over the sub-polar oceans, the large negative SWN anomalies exceed the positive TH and LWN
anomalies, resulting in an average of negative net SEB anomalies of -8 W m$^{-2}$. The largest negative impacts are located over
the Bering Sea, North Atlantic, and the western margins of Greenland extending to the Baffin Bay and Labrador Sea. The
dominance of SWN anomalies on the net SEB is due to the low albedo over the ocean regions. Other regions exhibit areas of
positive or weak negative net SEB AR impacts, with area average anomaly values ranging from 3 to 10 W m$^{-2}$. The larger
positive net SEB anomalies are found over the high albedo sea ice and ice sheet surfaces while smaller anomalies are seen
over the lower albedo continental areas due to the larger cooling effect from SWN in this region (Fig. 5b). Over the Arctic
Ocean, a distinct coastal contrast is evident, likely sustaining sea ice melt over the sea ice margins in summer.

While weak AR-induced absolute net SEB anomalies are observed over land, their total contributions to the mean net SEB
(Fig. 7c) are most evident over continents, particularly in cold seasons (spring: 90%, winter: 50%, fall: 24% in Table 1), with
much lower contribution in summer (3%), and Greenland throughout the year (>54%), far exceeding their corresponding AR
frequency, except over continental regions in summer. The large relative AR contribution to the net SEB is a result of the small
climatological mean net SEB over the continents and Greenland (Fig. 7a). The sub-polar oceans, which experience the largest
positive (downward) absolute impacts in cold seasons (Fig. 7b), contribute less in a relative sense to the mean SEB, ranging
from 65% in spring to 8%-9% in fall and winter due to the large climatological values of net SEB. In summer, ARs result in a
cooling impact on the net SEB with a relative contribution of -8%. In the central Arctic, the relative contributions are even
smaller, ranging from 32% in spring, 7-8% in fall and winter, and 1% in summer. This is attributed to the smaller magnitudes
in both AR occurrence frequency and AR-induced total net SEB anomalies. As a result, the AR relative contribution to the net
SEB is less than the AR frequency in all seasons, except spring, when it exceeds the AR frequency by 21%.

Focusing on the central Arctic the results discussed above that ARs make a negligible relative contribution to the net SEB in
all seasons except spring are surprising. Fig. 7b illustrates the substantial absolute impact of ARs on net SEB in winter,
averaging 39 W m$^{-2}$ (Table 1), with maximum anomalies extending to the Arctic Pacific and Atlantic sectors. This explains
the attention of Arctic AR literature on this season and area (e.g., Zhang et al., 2023b; Baggett et al., 2016). However, these



large anomalies occur with low frequencies (Fig. 1) such that when considering their cumulative impact on the net SEB they make small relative contributions that are less than the AR frequency in all seasons except spring. Nevertheless, local maximum

contributions to the mean SEB over the Arctic sea ice margins are observed, including the Bering-Chukchi Seas and Barents-Kara Seas in spring.

## 4 Discussion

Previous studies (Baggett et al., 2016; Fearon et al., 2021; Hegyi and Taylor, 2018; Woods and Caballero, 2016; Woods et al., 2013; Zhang et al., 2023b) have primarily focused on ARs, or analogous strong moisture intrusions, emphasizing their impacts

on LWD in specific case studies or limited geographic and seasonal context. In contrast, our study expands upon these earlier investigations by conducting a comprehensive assessment of the impact of Arctic ARs on all terms in the SEB. We explore AR-induced radiative and turbulent SEB flux average anomalies, their seasonal variation, and relative contributions to the mean SEB over a continuous 40-year period (from1980 to 2019) - a perspective that has not been adequately explored in existing literature. Both positive and negative anomalies in AR-induced SEB terms of the Arctic have climatological

significance, as they signify deviations in the energy budget at the surface, impacting local temperature, as well as influencing the rate of ice growth/melt (Persson et al., 2017; Serreze et al., 2007).

### 4.1 AR-induced surface and air temperature response

Our findings are mostly consistent with prior research, emphasizing that variations in Arctic surface temperature are predominantly driven by changes in LWD across various spatial and temporal scales (Cullather et al., 2016; Gong and Luo,

2017; Kim et al., 2017; Murto et al., 2023; Persson et al., 2017; Woods and Caballero, 2016). By comparing the spatial patterns presented in Figs. 2 and 4, our study confirms that the 40-year averaged LWD (Fig. 2a) and AR-related LWD anomalies (Fig. 2b) closely correspond to climatological mean surface and T-2m air temperatures (Fig. 4a, c) and AR-induced surface and air temperature anomalies (Fig. 4b, d), particularly in cold seasons (fall, winter, and spring).

We observe that the spatial patterns of AR-induced surface temperature and T-2m anomalies significantly differ from AR-induced SEB anomalies (Fig. 7b) but closely resemble AR-related LWD anomalies (Fig. 2b). Overall, ARs induce surface and near-surface atmospheric warming in the Arctic, particularly in cold seasons, with maximum warming in winter. This effect displays a clear land – sea – sea ice contrast, with amplified warming over land, to a lesser extent over the sea ice covered central Arctic Ocean, and a minimum impact over sub-polar oceans. On average, the AR-induced surface (T-2m) warming in

winter is largest over continents [9.5 K (9.4 K)] and Greenland [8.6 K (8.3 K)], less over the central Arctic [6.2 K (6.8 K)] and smallest over the sub-polar oceans [3.2 K (5.2 K)] (Table S2). In summer, the temperature responses exhibit different spatial patterns that do not resemble any of the AR-induced individual SEB component. Instead, the surface temperature response in summer (Fig. 4b) closely mirrors the AR net SEB anomalies (Fig. 7b): cooling responses over Arctic Pacific, Arctic Atlantic



- western Greenland – Baffin Island, and the North Pole are influenced by negative net SEB anomalies, while warming effects
over land are affected mainly by positive net SEB anomalies. The uniform weak warming observed over the sea ice covered
Arctic Ocean in summer contrasts with the amplified positive net SEB anomalies over the marginal sea ice area (Fig. 7b). This
difference is likely due to the constraint imposed by melting of sea ice, limiting the surface temperature increase. Consequently,
the average AR-induced surface (T2m air) response in summer is nearly 0 K (0.9 K) for sub-polar oceans, 0.1 K (0.8 K) for
the Arctic Ocean, and 1.1 K (1.7 K) and 2.8 K (2.9 K) for continents and Greenland, respectively (Table S2).


Furthermore, the surface and near-surface responses induced by ARs (Fig. 4b, d) strongly influence subsequent anomalies in
the SEB components. Specifically, during cold seasons, amplified surface and near-surface responses are observed. The large
AR-induced surface warming over land and Greenland results in stronger compensating LWU anomalies, consequently
yielding smaller LWN anomalies (Fig.3b), despite large LWD anomalies (Fig. 2b). Simultaneously, ARs produce comparable
surface and near-surface air temperature warming (Fig. 4d), resulting in small TH anomalies over land. In contrast, the limited
surface warming over sub-polar oceans gives rise to substantial LWN anomalies. This, along with the warmer air associated
with ARs (Fig. 4d), generate pronounced positive TH anomalies (Fig. 6b). Over the Arctic Ocean, moderate rises in surface
and near-surface temperatures, along with the related LWU anomalies associated with ARs, partially offset the large LWD
anomalies, resulting in the secondary large LWN and TH anomalies.


Moreover, Fig. 4e-f show that surface temperature inversions (shaded in blue in Fig. 4e) experience a slight decrease over the
sea ice covered central Arctic Ocean in winter (Fig. 4f). In contrast, a pronounced reduction in surface temperature inversions
is noted over a large part of high-elevation mountainous areas in continents during winter, with smaller scattered area in fall
and spring, and persistently in the ablation zone of Greenland throughout the year. In summary, during cold seasons, AR-
related surface and air temperature responses are mainly driven by LWD anomalies associated with ARs. While in summer,
the responses are influenced by anomalies associated with the net SEB, with the surface response closely resembling that of
the net SEB anomalies.

### 4.2 AR's crucial role in triggering Greenland Ice Sheet melt

Throughout the year, ARs exert significant warming effects on Greenland, with the most pronounced impacts occurring during
cold seasons (Fig.4b, d). Greenland, being one of the driest regions in the Arctic, makes it especially susceptible to AR impacts.
In cold seasons, the intensified surface warming in Greenland (7-8 K, as shown in Table S2) is primarily driven by strong
positive LWD anomalies associated with ARs (Fig. 2b), in contrast to the weaker positive TH anomalies (Fig. 6b). In contrast,
during summer, the moderate warming (3 K) over Greenland aligns more closely with positive SEB anomalies, suggesting
anomalies in each SEB component during ARs contribute to the surface response and subsequent melting effects.




Notably, in the southwest of Greenland where ARs are frequent (Fig.1), previous research (Mattingly et al., 2018) suggests that ARs contribute to substantial surface mass loss in summer and that this contribution is primarily attributed to TH anomalies, particularly enhanced SH anomalies. Mattingly et al., (2020) argue that the amplified SH anomalies may result from strong southerly winds at lower elevations, a significant temperature contrast between the ice surface and the near-ice

atmosphere associated with ARs and increased aerodynamic roughness in snow-free areas. This differs from our findings that show small TH anomalies (Fig. 6b) and larger LWN anomalies (Fig. 3b) that dominate the net SEB AR-induced anomalies (Fig. 6b) over Greenland in summer. This discrepancy with Mattingly et al., (2020) possibly results from their focus on the strongest AR days, where the maximum IVT exceeds the 90th percentile of all AR IVT at each basin and each season. At higher elevations in Northeast Greenland, certain distinct patterns emerge. AR-related LWN anomalies (Fig. 3b) are weakly positive,

while SWN anomalies (Fig. 5b) are weakly negative and the TH anomalies (Fig. 6b) dominate the net SEB anomaly in this region (Fig. 7b). These anomalies collectively contribute to the localized smaller positive net SEB anomalies in Northeast Greenland (Fig. 7b), leading to subsequent weak surface warming over the Northeast ablation zone (Fig. 4b, d).

### 4.3 Influence of AR detection methods on results

The characteristics of ARs and the consequent impact on SEB are significantly influenced by the methods used to detect ARs.

Different AR detection algorithms can yield varying AR statistics. For illustration, we compare the results with those from Ma et al., (2023, referred to as M23), which is based on the algorithm developed by Guan and Waliser, (2019) but applies it to ERA5 reanalysis, for AR occurrence frequency (Fig. 8a), AR LWD anomalies (Fig. 8b), and the overall contribution to mean SEB (Fig. 8c) for the same 40-year period analyzed above. Detailed information regarding the AR detection method of M23 is provided in Guan and Waliser, (2019) While both methods share common $IVT$-85th percentile climatological thresholds and

a 1500 km length criterion, M23 algorithm introduces additional requirements on the mean $IVT$ direction and coherence in $IVT$ directions. However, the most notable distinction between the two algorithms lies in the requirement of a minimum $IVT$ of 100 kg m$^{-1}$ s$^{-1}$ to complement the 85th percentile climatological thresholds in the M23 index.

In Fig. 8a, a significant year-round reduction in AR occurrence frequency across the entire Arctic region is evident when using

the M23 index, in contrast to the results shown in Fig. 1. Notably, during the cold seasons, particularly over the central Arctic Ocean, very few ARs are detected, as indicated by the large light blue area in Fig. 8a. The area-averaged AR occurrence frequency for the four regions range 1% to 7% in cold seasons (Table S3), in contrast to above 12% in the results discussed in Section 3.1. These distinct spatial differences between the two AR detection methods are primarily attributed to the minimum $IVT$ requirement of 100 kg m$^{-1}$ s$^{-1}$ used in the Guan and Waliser, (2019). As shown in Fig. S4, for January (representative of

winter), the 85th percentile climate threshold in most of the Arctic domain falls below 100 kg m$^{-1}$ s$^{-1}$. This results in minimal AR occurrences over the central Arctic Ocean in winter, as well as in fall and spring, in the M23 index (Fig. 8a). Consequently, any detected ARs will be associated with substantially higher $IVT$ values compared to those detected in the analysis.



In Fig. 8b, it is evident that M23 identified ARs consistently generate substantial positive LWD anomalies in the Arctic. In
winter, including only the most extreme AR events results in larger LWD anomalies over the central Arctic Ocean (73.4 W m$^{-2}$, in contrast to 45.1 W m$^{-2}$ in the results presented in the previous section), extending to both the Eurasian and North American continents compared to the results in Fig. 2b. These anomalies coincide with the lowest AR occurrence frequency in Fig. 8a, averaging 1.3 %, with maximum values exceeding 100 W m$^{-2}$. Similar to the analysis in Fig 2b, there are localized stronger AR-induced LWD anomalies over the sea ice margins of the central Arctic Ocean, particularly from the Greenland Sea to the
Barents-Kara Seas in winter. The M23 results also show larger AR LWD anomalies in spring (41-62 W m$^{-2}$ for all 4 regions in Table S3), compared to Fig. 2b (33-44 W m$^{-2}$ for all 4 regions in Table 1), but similar anomalies in summer and fall, suggesting that the stricter AR detection threshold does not make much difference in these two seasons when the climatological value of IVT is larger.

Although M23 identified ARs have larger absolute magnitudes in AR-related LWD anomalies in winter and spring, their relative contributions to the mean SEB are smaller (Fig. 8c), due to the lower AR frequency of occurrence (Fig. 8a). In fact, the relative contribution of AR-related LWD anomalies is consistently less using the M23 AR index, compared to the results shown in Fig. 2c in all seasons, although the spatial pattern of the AR contribution is similar for both AR detection methods. The largest contribution from the M23 remains in spring, but their magnitudes are much smaller, with an average of 7% over
Arctic Ocean (in contrast to 45% in Table 1), 40% over sub-polar oceans (in contrast to 69% in Table 1), 79% and >300% over continents and Greenland (in contrast to >200% and >4000% in Table 1). Central Arctic regions receive minimal AR-induced LWD contributions to the mean SEB, ranging from 7% in spring (45% in Table 1) to 1-2% in summer, fall, and winter (2-10% in Table 1).

The use of a 100 kg m$^{-1}$ s$^{-1}$ threshold is a common practice in Arctic AR studies (Fearon et al., 2021; Guan and Waliser, 2015, 2019; Ma et al., 2020; Zhang et al., 2023b, and others), originally chosen to distinguish AR-like features in polar regions and to align with the AR detection results for East Antarctica by Gorodetskaya et al., (2014), which captured extreme precipitation events. Results from the M23 index indicate that using a 100 kg m$^{-1}$ s$^{-1}$ threshold detects much fewer AR occurrences and yields minimal contributions to the mean SEB over the central Arctic Ocean in winter, despite the presence of larger absolute
LWD anomalies. Even in the Barents-Kara Seas, where ARs have been recently examined for significant LWD anomalies, the maximum contribution of AR-related total LWD anomalies is less than 4%, even lower than the corresponding AR frequency of less than 5% in winter. Contrary to the findings in Section 3.2.1, the M23 index indicates that over the entire 40-year period, AR-related total LWD anomalies contribute minimally to the mean SEB and have a limited impact on delaying sea ice refreeze in winter. This suggests that the use of restrictive criteria for AR detection emphasizes impacts for individual cases but results
in minimizing the contributions by the ARs to the overall climatology of the SEB due to their very low frequency of occurrence. The initial comparison of two AR detection methods presented here underscores the need to assess various Arctic AR detection tools and establish a comprehensive Arctic AR intercomparison project, which will significantly contribute to advancing our



knowledge of Arctic ARs and their effects, including SEB effects. Ultimately, such efforts will help address the uncertainties regarding AR impacts due to the use of different AR detection methods.








**Figure 8. According to the M23 AR index, spatial distributions of (a) 40-year AR occurrence frequency (percentage of AR occurrence frequency) at each of the four seasons: Spring (March-May), Summer (June-August), Fall (September-November) and Winter (December-February) from 1980 to 2019. The black lines, ranging from thin to thick, represent AR frequencies at 1%, 3%, 5%, and 7%, respectively. (b) 40-year mean LWD anomalies (unit: W m$^{-2}$) during the presence of AR events at each of the four seasons. Positive values indicate downward. (c) the fraction of 40-year AR contribution (unit: percent) from the total LWD anomalies to the absolute values of 40-year average net SEB during each of the four seasons. The percentage results greater than 100% are shaded in grey for clarity. The blank areas in the figures correspond to areas with zero AR occurrence frequency. Fig. 8 (a), (b), and (c) are similar to Fig.1, Fig 2(b), and Fig. 2(c), respectively, but for the results according to M23 AR detection index.**

### 4.4 Limitations of the reanalysis data

The results presented here relies on reanalysis data, used due to constraints from the limited Arctic observations compared to mid-latitudes. While ERA5 exhibits superior performance among reanalysis datasets in the Arctic (Graham et al., 2019b), it remains dependent on climate models and data assimilation skills, potentially deviating from actual conditions. Notably, ERA5 shows biases in T2m during winter and spring, leading to poorly represented surface inversions and turbulent heat fluxes over sea ice (Graham et al., 2019a; Herrmannsdörfer et al., 2023). Further case studies examining Arctic AR events and their SEB impacts should compare different reanalyses as well as utilize observational data, such as from the MOSAiC expedition (Shupe et al., 2022), which can not only validate the ERA5 reanalysis but also contribute to a more accurate and comprehensive understanding of AR impacts in the Arctic region.

### 5 Conclusions

In this study, we focus on Arctic AR events identified by using IVT exceeding the 85th climatological percentile in 3-hourly ERA5 reanalysis data from January 1980 to December 2019. We conduct a comprehensive analysis of the SEB components during AR events by examining their absolute average anomalies (panel b in Figs. 2-3, 5-7) and overall AR contribution to seasonal means of SEB across the Arctic region (panel c in Figs. 2-3, 5-7). Our findings reveal substantial variations in SEB components and net SEB during AR events, characterized by distinct seasonality and pronounced land - sea - sea ice contrast patterns.

Over sea ice-covered central Arctic Ocean, ARs produce substantial positive LWD anomalies (Fig.2b), especially in cold seasons (29-45 W m$^{-2}$), with the most notable effects over marginal sea ice areas. The moderate surface warming induced by AR LWD leads to corresponding moderate LWU anomalies, partially offsetting the large LWD anomalies and creating moderate impacts on LWN anomalies (16-22 W m$^{-2}$). Combined with similar magnitudes of the TH anomalies (15-21 W m$^{-2}$) and weak negative SWN anomalies, large positive net SEB impacts (26-39 W m$^{-2}$) are observed. These impacts potentially hinder sea ice refreezing in fall and winter and may trigger sea ice melt in spring. In summer, more pronounced AR SWN anomalies (-22 W m$^{-2}$) largely counteract the LWN anomalies (14 W m$^{-2}$) and TH anomalies (17 W m$^{-2}$), resulting in a weak



positive net SEB impact (9 W m$^{-2}$). However, local maximum positive net SEB anomalies are observed over sea ice margins, potentially accelerating sea ice melt in summer (Fig. 7b). When normalized by the mean SEB in Fig. 7a and considering the

AR occurrence frequency in Fig.1, spring stands out as the season with the most significant AR-induced relative contribution to the mean SEB (Figs. 2c, 3c, 5c, 6c, and 7c). On average, AR-induced total LWD anomalies contribute 45% to the mean SEB in spring, which potentially serve to initiate sea ice melt (Huang, Dong, Bailey, et al., 2019; Huang, Dong, Xi, et al., 2019). Moreover, AR-related total net SEB anomalies contribute 31.8% (Table 1) to the mean SEB, exceeding their corresponding occurrence frequency (10.8%), which potentially initiate the spring sea ice melt and may influence the minimum

sea ice extent in fall (Huang et al., 2019). However, ARs make a small relative contribution to the net SEB in other seasons, lower than the corresponding AR frequency, indicating that ARs play a less important role compared to their frequency.

Over sub-polar oceans, ARs exert the most significant absolute positive impact on net SEB in cold seasons (40-91 W m$^{-2}$), which is primarily attributed to substantial positive TH anomalies (42-62 W m$^{-2}$) and, to a lesser extent, positive LWN

anomalies (16-22 W m$^{-2}$) associated with a weak surface temperature response. However, the overall contribution to the mean SEB is most significant in spring (Fig. 7c), averaging 65.3% as seen in Table 1. In summer, strong negative SWN anomalies dominate (-52 W m$^{-2}$) due to the low albedo over the oceans, leading to overall negative net SEB anomalies (-8 W m$^{-2}$) and associated weak negative effects on the mean SEB (-1%).

ARs generate relatively smaller absolute anomalies in net SEB over continental areas (Fig. 7b), mainly attributable to the LWN. However, their impact on the mean SEB is substantial, especially in cold seasons (24-90%), far exceeding the corresponding frequency, which is primarily due to the smaller mean SEB (Fig. 7a). The large relative contribution underscores AR's crucial role in determining the net SEB over continents, except in summer (3%, lower than their frequency).

Greenland, particularly western Greenland, exhibits noteworthy AR-related LWD absolute anomalies (34-47 W m$^{-2}$) and a significant relative contribution to mean SEB (> 100%) throughout the year. These substantial AR LWD anomalies largely drive the net SEB anomalies (10-28 W m$^{-2}$) and contribute significantly to mean SEB (> 54%). In summer, AR-related total net SEB anomalies contribute to 62.5% of the mean SEB, a six-fold increase compared to the corresponding AR occurrence frequency (11.1%). The amplified contribution has the potential to trigger melt over the Greenland Ice Sheet (Mattingly et al.,

2018, 2023, 2020; Neff, 2018; Neff et al., 2014).

In summary, our work underscores the crucial role of ARs in influencing absolute SEB anomalies, especially during cold seasons, particularly winter, which has traditionally received the greatest attention in AR research. However, when considering AR occurrence frequency, we find that the most significant contribution to the mean SEB occurs during spring. This effect is

particularly notable over the sea ice-covered central Arctic Ocean and may have profound implications for sea ice thermodynamics in marginal and pack ice regions. Our research enhances our understanding of Arctic warming and sea ice



decline within the context of ongoing Arctic amplification. Furthermore, these findings hold the potential to enhance climate models, leading to more accurate predictions of future Arctic climate changes and informed decision-making to mitigate the impacts of Arctic amplification at regional and global scales.


*Code and data availability.* ERA5 data were acquired from the Copernicus Climate Change Service (C3S) Climate Data Store (https://cds.climate.copernicus.eu/cdsapp#!/dataset/reanalysis-era5-single-levels?tab=form).

The AR index crucial for supporting our conclusions is openly accessible after an embargo period at the following URL: https://purr.purdue.edu/publications/4322/1 (Tung et al., 2023). The Ma23 AR index and all code necessary to replicate the

presented results will be made available upon request from the corresponding author, Chen Zhang (chen.zhang-3@colorado.edu)

*Author contributions.* Zhang, Cassano, and Seefeldt conceived the study and devised the methods. Zhang curated the data, conducted the analysis, visualized the results, and wrote the original manuscript all under the supervision and support of

Cassano and Seefeldt. Zhang, Tung, Wang, and Ma facilitated access to the atmospheric river detection algorithm. All authors actively contributed to the discussion and participated in reviewing and editing the manuscript.

*Competing interests.* The contact author has declared that none of the authors has any competing interests.

*Acknowledgement.* Zhang is supported by the Cooperative Institute for Research in Environmental Sciences (CIRES) Visiting Fellows Program, funded by the National Oceanographic and Atmospheric Administration (NOAA) Cooperative Agreement NA22OAR4320151. Cassano, Seefeldt, Wang, and Ma are supported by the Regional and Global Model Analysis (RGMA) component of the Earth and Environmental System Modeling (EESM) program of the U.S. Department of Energy's Office of Science, as a contribution to the HiLAT-RASM project. Tung is support by NSF OAC-2232872. We also acknowledge Purdue

Information Technology at Purdue (ITAP) Rosen Center for Advanced Computing (RCAC) for their generous assistance with data-intensive high-performance computing. We express our gratitude to Ola Persson, Matthew Shupe, Mark Serreze, and Patrick Taylor for their insightful comments and engaging discussions, which significantly contribute to enhancing the quality of our presentation. Additionally, we thank Qin Kong for facilitating access to the ERA5 data.

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
