# Peer review of "Quantifying the Impacts of Atmospheric Rivers on the Surface Energy Budget of the Arctic Based on Reanalysis"

_EGUsphere, 2024_

## Referee Comment (RC3)

Review of "*Quantifying the Impacts of Atmospheric Rivers on the Surface Energy Budget of the Arctic Based on Reanalysis*" by Zhang et al.

In this paper, authors aim to analyze the contribution of atmospheric rivers (ARs) to the seasonal surface energy budget (SEB) in the Arctic using ERA5 reanalysis data for 1980-2019. ARs are detected using the 85[th] percentile of IVT and components of the seasonal SEB are anomalies are assessed for times when ARs are detected. The aim of improving understanding the importance of ARs in net SEB in the Arctic is important and interesting, and the authors provide a very detailed analysis with discussion of implications and connections to previous work. Analysis regarding absolute anomalies is thorough, but I am unsure of the appropriateness of the metric used to quantify the contributions of ARs to seasonal SEB (detailed in Major Comment 1).

**Major Comments:**
1. The metric used for evaluating the contribution of ARs to net SEB may not be appropriate for the conclusions drawn. It is difficult to interpret the physical meaning of the contributions when the seasonal net SEB is very small. Based on the description of the calculation, it appears that the contribution is being calculated as (using downwelling longwave (LWD) as an example):

$$\frac{(LWD_{AR} - LWD_{All}) * \left(\frac{t_{AR}}{t_{All}}\right)}{|SEB_{All}|} = \frac{(LWD_{AR} - LWD_{All}) * t_{AR}}{|SEB_{All}| * t_{All}} \quad \text{(Equation 1)}$$

where $t_{AR}$ and $t_{All}$ represent the number of 3-hr time intervals when ARs are present and the total number of 3-hr time intervals in the season, respectively. The physical meaning of that quantity is unclear because of how the times are used to scale the flux ratio.

Considering the net SEB to be
$$SEB = LWD - LWU + SWD - SWU + SH + LH \quad \text{(Equation 2)}$$

where $LWU$ and $SWU$ are upwelling long- and shortwave radiation, respectively, and $SH$ and $LH$ represent the sensible and latent heat fluxes (assuming turbulent fluxes are positive downwards). It's quite possible for the total balance to nearly cancel each other out (as seen in Greenland in Spring, Table 1). This means that comparing, for example, LWD_{AR} − LWD_{All} to |SEB_{All}| could yield very large percentages (over 1000%) even if the difference between LWD_{AR} and LWD_{All} is much less than 100% (e.g., if SEB_{All} is 0.4 W m^{-2}, LWD_{All} is 177.2 W m^{-2}, and LWD_{AR} is 221.0 W m^{-2}). In this case, the % difference for LWD is only about 25%, but Equation 1 suggests a results of 1292%. That is not physically meaningful.

To yield more physically meaningful results, the anomaly for each component (F) can be compared to the mean values of that component;
$$\frac{(F_{AR} * t_{AR}) - (F_{All} * t_{All})}{(F_{All} * t_{All})} \quad \text{(Equation 3)}$$
where F may represent any SEB component or net SEB. This would provide an estimate of the magnitude of the anomaly relative to the average value for each component. Using this option instead of the current construction would be clearer and more easily justified.

However, since a main goal of this paper is to estimate the relative contribution of different components to net SEB, the net SEB can be decomposed for each flux relative to SEB by looking at the total energy ($E_{tot}$) over the season as the sum of the net SEB during ARs and net SEB during times without ARs;

$$E_{Tot} = SEB_{All} * t_{All} =$$
$$(LWD_{AR} - LWU_{AR} + SWD_{AR} - SWU_{AR} + SH_{AR} + LH_{AR}) * t_{AR} +$$
$$(LWD_{NoAR} - LWU_{NoAR} + SWD_{NoAR} - SWU_{NoAR} + SH_{NoAR} + LH_{NoAR}) * (t_{All} - t_{AR})$$
(Equation 4)

Then, replacing a single term in the AR portion of the equation with a NoAR value in a hypothetical calculation ($E_{Hyp}$) can provide an estimate of the difference due to that single term. Using LWD as an example:

$$E_{Hyp} =$$
$$(LWD_{NoAR} - LWU_{AR} + SWD_{AR} - SWU_{AR} + SH_{AR} + LH_{AR}) * t_{AR} +$$
$$(LWD_{NoAR} - LWU_{NoAR} + SWD_{NoAR} - SWU_{NoAR} + SH_{NoAR} + LH_{NoAR}) * (t_{All} - t_{AR})$$
(Equation 5)

The difference between $E_{Tot}$ and $E_{Hyp}$ can tell you the absolute impact of LWD during ARs, and the relative impact can be found by dividing by $E_{Tot}$:

$$\frac{E_{Hyp} - E_{Tot}}{E_{Tot}}$$     (Equation 6)

This is another potential solution that would result in more physically meaningful results in representing the ARs' contribution to the seasonal energy budget that might be preferable to the authors since it maintains an overall energy value as the denominator. However, note that there is no simultaneous changing of the flux and the amount of time here.

Regardless of how the authors proceed, the equation used to calculate this metric should be included, rather than only described in words to make sure it is very clear what is being shown.

2.  I am unable to reproduce the "contribution to SEB" values shown in Table 1 using the description of how it was calculated in Section 2.3. Since the AR frequencies, anomalies and net SEB values are provided for each region, the contribution should be able to be calculated without any further information (based on Section 2.3). For example, I get the following for spring in the Central Arctic:

$$\frac{(LWD_{AR} - LWD_{All}) * \left(\frac{t_{AR}}{t_{All}}\right)}{|SEB_{All}|} = \frac{(32.9\ W\ m^{-2}) * (0.108)}{|-19.6\ W\ m^{-2}|} = 0.181 = 18.1\%$$

(Equation 7)

Whereas Table 1 shows 45.1%. Please ensure methods are described clearly so the results can be reproduced.

3. Consider performing statistical testing to determine if the absolute anomalies during ARs are statistically different from the mean conditions (which could be shown in the b rows of Figures 2-7). Since ARs exist in a location likely for more than one timestep, there is some temporal autocorrelation which may be accounted for by randomly selecting a smaller sample of AR timesteps to compare to a randomly selected sample of non-AR timesteps. Determining the statistical significance of these anomalies may help to identify SEB components that are more important with more confidence.

**Minor Comments:**

- 45-46: ARs typically being associated with extratropical cyclones is mentioned here, but isn't discuss it again. I think more discussion regarding the linkage between cyclones and ARs would be valuable here for context of when/how ARs occur in the Arctic.

- 100: It is mentioned that times are only used during neutral or weak El Niño-Southern Oscillation. I assume it's because of IVT anomalies associated with strong ENSO events, but it is worth briefly stating in the text for clarity.

- 123-125: Is it necessary to give multiple names for these first 3 ERA5 variables?

- 147-149: This sentence uses both "three-hourly" and "3-hourly" referring to the data – I suggest picking one to remain consistent.

- 287-289: What is meant by "ARs make their most significant relative contribution to the average net SEB in spring, accounting for at least 45% of the net SEB, surpassing the corresponding AR frequency by more than 34%"? I don't think subtracting the frequency from the contribution has a physical meaning since they are percentages of different things.

- 358: I suggest starting a new paragraph at "The results over the central Arctic" as this is a long paragraph, and a new topic is being introduced here.

- Section 3 is titled "Analysis and Results" and Section 4 "Discussion", but Section 3 includes a lot of discussion (i.e., discussing potential impacts of the anomalies, comparing to previous work) and Section 4 still discusses some results (particularly temperature). A potential solution for this would be to rename Section 3 to focus on SEB and Section 4 to focus on impacts, and perhaps create another section for limitations/uncertainties (for 4.3 and 4.4).

---

## Author Comment (AC1)

**egusphere-2024-320**
**"Quantifying the Impacts of Atmospheric Rivers on the Surface Energy Budget of the Arctic Based on Reanalysis"**
**Response to the Reviewers**

**By Chen Zhang, John J. Cassano, Mark Seefeldt, Hailong Wang, Weiming Ma, and Wenwen Tung**

We appreciate the valuable comments provided by the Reviewers. Before addressing each point individually, we would like to acknowledge the two common concerns raised by Reviewers.

Firstly, there were concerns regarding the methodology of our analysis. The primary objective of this work is to estimate the relative contribution of different surface energy budget (SEB) components to the net SEB. To achieve this, original panel (c) in the Figures 2-3, 5-7 of the manuscript aims to illustrate the relative AR contribution to SEB components, normalized by the net SEB. This normalization involves calculating the ratio of the accumulated AR SEB term, which accounts for both the magnitude of individual AR anomalies and their frequency of occurrence, to the accumulated seasonal net SEB. By adopting this normalization approach, we enable consistent comparisons across different SEB components, thereby allowing readers to discern relative contributions effectively.

Furthermore, following RC3's suggestion with a slight modification, we chose to calculate the relative contribution of AR-related SEB component anomaly normalized by the mean of each respective component. This approach aims to estimate the accumulated AR contribution of SEB component relative to their total values. We chose to present the results as an additional panel, now labeled as new panel (c), in Figures 2-3, and 5-7 of the revised manuscript. Consequently, the original panel (c), depicting the AR SEB contribution normalized by the total SEB, has been reassigned to panel (d) to accommodate this adjustment.

Specifically, the results shown in new panel (c) result from the following calculation at each individual grid point within the study domain for each season:

1. Calculate the total extra energy contributed by each SEB component when ARs are present as, $(F_{AR} - F_{All}) * t_{AR}$, where $F_{AR}$ represents the mean of any term in the SEB equation when an AR is present, $F_{All}$ denotes the seasonal mean of any term in the SEB equation, and $t_{AR}$ indicates the total number of 3-hourly time steps during which ARs are present.

2. Calculate the total energy for each component as, $F_{All} * t_{All}$, where $t_{All}$ signifies the total number of 3-hourly time steps within each season.

3. Determine the ratio of these two terms, which provides an estimate of the magnitude of AR anomaly for each SEB term relative to the average value for each component. This is

presented in Eq. (2) in the manuscript, noting the ratio of $t_{AR}$ to $t_{All}$ is simply the AR frequency shown in Fig. 1

$$\frac{(F_{AR}-F_{All})* t_{AR}}{F_{All}*t_{All}} = \frac{panel\ (b)* Fig.1}{panel\ (a)} \tag{2}$$

Additionally, we include the net SEB equation in the revised manuscript, labeled as Eq. (1), as follows:

$$net\ SEB = LWN + SWN + TH = LWD - LWU + SWD - SWU + SH + LH \tag{1}$$

Where LWN, SWN and TH denote the net longwave radiation, net shortwave radiation, and turbulent heat flux, respectively. LWD, LWU, SWD, SWU, SH, LH represent downward longwave, upward longwave, downward shortwave, upward shortwave, sensible and latent heat flux, respectively.

Secondly, two Reviewers expressed concerns about the organization of our sections, particularly noting overlapping discussions between Section 3 (Analysis and Results) and Section 4 (Discussion). To address this issue, we have restructured the sections as follows:
- Section 3: AR occurrence frequency (original Section 3.1)
- Section 4: AR's influence on the surface energy budget component of the Arctic (original Section 3.2)
   - Section 4.1: Surface radiative fluxes (original Section 3.2.1)
      - Section 4.1.1: Surface downward longwave radiation
      - Section 4.1.2: Net surface longwave radiation
      - Section 4.1.3: Net surface shortwave radiation
   - Section 4.2: Surface turbulent heat fluxes (original Section 3.2.2)
   - Section 4.3: Net Surface energy budget (original Section 3.2.3)
- Section 5: AR's surface impacts
   - Section 5.1: AR-induced surface and air temperature response (original Section 4.1)
   - Section 5.2: AR's crucial role in triggering Greenland Ice Sheet melt (original Section 4.2)
- Section 6: Uncertainties and limitations
   - Section 6.1: Influence of AR detection methods on results (original Section 4.3)
   - Section 6.2: Limitation of the reanalysis data (original Section 4.4)
- Section 7: Conclusions (original Section 5)

We believe these adjustments will enhance the clarity and coherence of our manuscript, addressing the concerns raised by the Reviewers effectively.

Below, we respond in blue text to the Reviewer's comments, using an italic font to indicate text that has been copied verbatim from the Reviewer's reports.
* * *
Reply to RC1, Jeff Ridley:
We appreciate the Reviewer for the valuable criticisms and constructive comments. We are particularly grateful to the Reviewer for suggesting interesting avenues for future research. Regrettably, due to the journal length limitation, we cannot incorporate every suggestion. However, we assure the Reviewer that we have carefully considered each recommendation and integrated those feasible within the scope of our paper. Regarding the concern about the methodology of our analysis, we have provided a detailed explanation below.

***RC1, Jeff Ridley:***
*The methodology of this paper is flawed. Not only are the atmospheric rivers (AR) included in the climatologies used, and thus cannot exceed 100% of the budgets, but the local fluxes within the bounds of the AR are calculated as an anomaly without consideration of the budget for region as a whole i.e. reflecting the fractional area of the AR to the area of the region as a whole (e.g. Greenland, marginal seas etc.*
Reply: We appreciate the Reviewer's insightful comments and apologize for any lack of clarity in our methodology section. Below, we describe in detail our methodology to attempt to alleviate any confusion, although we are unsure of what the Reviewer is suggesting with the comment "*without consideration of the budget for region as a whole*". We will be happy to address further comments in a subsequent review if our explanation below is not sufficient.

The primary objective of our work is to assess the relative impacts of AR on various surface energy budget (SEB) components as shown in Figures 2, 3 and 5-7. To achieve this we calculate, on a grid point basis, the average SEB terms when ARs are present and compare this to the grid point mean for each term (panel a). The AR anomaly is the difference between the AR mean SEB term and the overall mean of that term (panel b). To quantify the contribution of the AR SEB to the overall SEB we compare the seasonal total of each SEB term during AR events to the total SEB (original panel c). Thus, original panel c illustrates the relative AR contribution to SEB components, normalized by the absolute net SEB. This normalization involves calculating the ratio of the accumulated AR SEB term, which accounts for both the magnitude of individual AR anomalies and their frequency of occurrence, to the accumulated seasonal net SEB.

A relative AR SEB contribution exceeding 100% indicates that the considered term has a greater AR contribution than the total SEB, implying that other SEB terms counteract to yield a small net SEB. We do not agree with the Reviewer that values greater than 100% are not possible or lack meaning. If we consider just the mean SEB the contribution of downward longwave radiation in winter will exceed the overall mean SEB because other terms in the SEB oppose the energy gain from downwelling longwave radiation, namely outgoing longwave radiation. Similarly for our AR results, values greater than 100% simply indicate that that term is contributing more energy than the total SEB and thus other terms in the SEB must oppose it. Further, very large, normalized values indicate that the overall SEB is the result of large,

oppositely signed terms and that the AR term being considered is one of those large terms. This normalization facilitates consistent comparison across different SEB components, allowing readers to discern relative contributions effectively.

Additionally, following another Reviewer's suggestion, we chose to calculate the relative contribution of AR-related SEB component normalized by the mean of each respective component, shown in the Equation (2). This approach aims to estimate the accumulated AR contribution of each SEB component relative to their total values. We chose to present the results as an additional panel, now panel (c), in Figures 2-3, and 5-7. Consequently, we have reassigned the original panel (c), the AR SEB contribution normalized by the total SEB, to panel (d) to accommodate these results.

To summarize the revisions made in the manuscript, panel (a) presents the climatology of SEB component. The inclusion of panel (b), depicting composite absolute AR-related SEB term anomalies adjacent to panel (c) and panel (d), which now respectively display the relative AR contribution to the average value for each component and total net SEB. By presenting both the anomaly (panel (b)) and relative contribution (panel (c) and panel (d)), we aim to provide readers a comprehensive perspective, highlighting terms that are large in both absolute and relative senses (e.g., downward longwave radiation over sea ice-covered central Arctic Ocean), as well as those that, despite small absolute anomalies, are substantial relative to the overall surface energy budget (e.g., SEB terms over continents).

Furthermore, we have included the equations used to calculate these results of panel (c) and (d) in Section 2 (Data and Methods) for transparency and clarity in the manuscript, as follows:

*"Mathematically, the results shown in panel (c) result from the following calculation at each individual grid point within the study domain for each season:*

1. *Calculate the total extra energy contributed by each SEB component when ARs are present as, $(F_{AR} - F_{All}) * t_{AR}$, where $F_{AR}$ represents the mean of any term in the SEB equation when an AR is present, $F_{All}$ denotes the seasonal mean of any term in the SEB equation (panel (a)), and $t_{AR}$ indicates the total number of 3-hourly time steps during which ARs are present.*
2. *Calculate the total energy for each component as, $F_{All} * t_{All}$, where $t_{All}$ signifies the total number of 3-hourly time steps within each season.*
3. *Determine the ratio of these two terms, which provides an estimate of the magnitude of AR anomaly for each SEB term relative to the average value for each component. This is presented in Eq. (2), noting that the ratio of $t_{AR}$ to $t_{All}$ is simply the AR frequency shown in Fig. 1.*

$$\frac{(F_{AR} - F_{All}) * t_{AR}}{F_{All} * t_{All}} = \frac{panel\ (b) * Fig.1}{panel\ (a)} \tag{2}$$

*Furthermore, the results depicted in panel (d) stem from the following calculation conducted at each individual grid point within the study domain for each season.:*

1. *Calculate the total extra energy contributed by each term in the SEB equation when ARs are present as: $(F_{AR} - F_{All}) * t_{AR}$*
2. *Compute the absolute value of total SEB energy as: $|netSEB_{All}| * t_{All}$, where $|netSEB_{All}|$ represents the absolute value of seasonal mean net SEB at a given grid point.*
3. *The ratio of these two terms indicates the relative contribution of the AR anomaly for each SEB term to the total seasonal SEB, as shown in Eq (3).*

$$\frac{(F_{AR}-F_{All}) * t_{AR}}{|net\ SEB_{All}|*t_{All}} = \frac{panel\ (b)*Fig.1}{|Fig.7(a)|} \qquad\qquad (3)"$$

As the reviewer comment suggests there is also a value in considering the AR impacts on a regional basis, and this is done for each SEB term in Table 1 and Table S1. These tables summarize AR occurrence frequency, climatological mean of each SEB term, composite AR anomalies for each SEB term, total AR contribution to individual SEB component, total AR contribution to absolute net SEB (AR anomaly times the time when ARs are present), and relative AR contribution to net SEB compared to the AR frequency across four regions. These results are derived from area-averaged calculations. Which involves summing the results of grid points falling within each region and weighting them using the cosine values of their corresponding latitudes. This approach ensures a representative assessment of AR impact across different regions. We have included the methods to calculate the results in Section 2.3 for clarity, as follows:

*"We summarize key features from Figures 2-3,5-7 into Table 1 and Table S1 to analyze each SEB component and the net SEB across four sub-regions: the central Arctic (including the Barents and Kara Seas), sub-polar oceans, continents, and Greenland (Fig. S1), for every season. These tables present regional averages for several metrics, including climatology (panels a), composite anomalies (panels b), AR contribution to individual SEB component (panels c), AR contribution to absolute net SEB (panels d), along with AR frequency (as shown in Fig.1). To derive these results, we perform area-averaged calculations by summing the values from grid points within each region and weighting them based on the cosine values of their corresponding latitudes. Additionally, we calculate the difference between the area-averaged AR contribution to the net SEB and the area-averaged AR frequency, representing additional AR contribution, which is presented in the last row of the tables."*

*Additionally, the authors to not make the case for AR vs extratropical cyclones. AR are not a standalone feature and thus the tropical cyclone itself is the story not the AR.*
Reply: Indeed, ARs are not a standalone feature and are always associated with a low-level jet and extratropical cyclone (according to Ralph et al., 2018: Defining "Atmospheric River": how

the glossary of meteorology helped resolve a debate). We also agree with the reviewer that performing an analysis similar to what we present in this manuscript for extratropical cyclones would be a worthy future research direction. However, the research community does consider assessing the contribution of solely ARs to be a relevant research topic as indicated by the numerous references cited in the manuscript.

Previous studies have predominantly focused on the individual impacts of ARs, emphasizing their roles in enhancing moisture, downward infrared radiation, and the consequent surface energy budgets (SEB) in specific contexts, such as case studies or limited geographic and seasonal domains (e.g., Hegyi and Taylor, 2018; Mattingly et al., 2018, 2023, 2020; Zhang et al., 2023). These existing literatures have motivated us to build upon their findings and undertake a comprehensive assessment of AR impacts on the SEB. Thus, we believe that following previously published AR studies, there is an interest within the research community to simply assess the role of ARs separate of any other associated features such as extratropical cyclones.

*Some other line by line points*
*Line 46. The argument here is that atmospheric rivers are a distinct feature when they are simply associated with extra-tropical cyclones. It is the cloud associated with the cyclone warm front that is leads to the excessive LW-down. The detrainment of water vapor from the cyclone could be adding to LW-down, but the authors are not distinguishing the two characteristics here. Include further references to add to Ralph et al., 2018 to show that there is considerable mechanistic literature on the cause of 'atmospheric rivers'.*

*Eiras-Barca, J., Ramos, A. M., Pinto, J. G., Trigo, R. M., Liberato, M. L. R., and Miguez-Macho, G.: The concurrence of atmospheric rivers and explosive cyclogenesis in the North Atlantic and North Pacific basins, Earth Syst. Dynam., 9, 91–102, https://doi.org/10.5194/esd-9-91-2018, 2018.*

*Zhang, Z., Ralph, F. M., & Zheng, M. (2019). The relationship between extratropical cyclone strength and atmospheric river intensity and position. Geophysical Research Letters, 46, 1814–1823. https://doi.org/10.1029/2018GL079071*

*Dacre, H. F., P. A. Clark, O. Martinez-Alvarado, M. A. Stringer, and D. A. Lavers, 2015: How Do Atmospheric Rivers Form?. Bull. Amer. Meteor. Soc., 96, 1243–1255, https://doi.org/10.1175/BAMS-D-14-00031.1.*
Reply: We have incorporated discussions on the linkage between cyclones and ARs into the manuscript, along with citations to the recommended literature, as follows:

*"In mid-latitudes, ARs are commonly identified in the warm conveyor belts of synoptic-scale cyclones, particularly low-level jets (Ralph et al., 2004, 2006). Some literature even considers*

*ARs as part of cyclones (Bao et al., 2006; Neiman et al., 2008; Dacre et al., 2015). ARs and cyclones exhibit strong statistical and dynamic relationships (Zhang et al., 2019; Guo et al., 2020; Eiras-Barca et al., 2018). In the Arctic, poleward moisture transport is also closely linked to cyclone activity, including intensity, frequency, and duration (Villamil-Otero et al., 2018). Arctic cyclones account for over 70% of the average annual moisture transport, with their track orientation and upper-level steering flow significantly influencing poleward moisture flux (Fearon et al., 2021)."*

*If you accept that 'atmospheric rivers' are manifestations of subtropical cyclones, as the above papers suggest, then reference to previous Arctic budget analysis is required.*

*Villamil-Otero, G.A., Zhang, J., He, J. et al. Role of extratropical cyclones in the recently observed increase in poleward moisture transport into the Arctic Ocean. Adv. Atmos. Sci. 35, 85–94 (2018). https://doi.org/10.1007/s00376-017-7116-0*
Reply: We have incorporated this literature you provided into the manuscript (as demonstrated in the above passage).

*Line 68. In any estimation of energy budget on needs to calculate the impact of snowfall associated with the cyclones on sea ice and land energy budgets, because of the high albedo of snow in spring.*
*Webster, M.A., Parker, C., Boisvert, L. et al. The role of cyclone activity in snow accumulation on Arctic sea ice. Nat Commun 10, 5285 (2019). https://doi.org/10.1038/s41467-019-13299-8*
Reply: Indeed, our findings did uncover distinct responses to AR SEB impacts across surfaces with varying albedos, as discussed in the manuscript's section on surface shortwave radiation associated with ARs. Specifically, we observed larger AR-related net surface shortwave radiation anomalies in lower albedo subpolar regions, contrasting with lower anomalies in the high albedo central Arctic Ocean and Greenland. However, as stated earlier, the primary objective of this study is to conduct a comprehensive examination of the impact of ARs on SEB impacts. While we acknowledge the insightful findings regarding cyclone activity and snow accumulation on Arctic sea ice from the study by Webster et al., 2019, it lies beyond the scope of our current focus. We have incorporated this discussion in the Section 6.1(original Section 4.3) and cited accordingly, shown below:

*"In addition, Arctic ARs are closely linked with Arctic cyclones, which strongly influence surface heat fluxes, particularly TH (Blanchard-Wrigglesworth et al., 2022), subsequently impacting the net SEB. Moreover, studies suggest that large SEB anomaly events in the Arctic are often associated with an increased frequency of cyclone occurrence (Murto et al., 2023). Additionally, cyclones affect snowfall accumulation on sea ice, thereby influencing SEB due to high albedo of snow (Webster et al., 2019). Our findings indicate that surfaces with varying albedos exhibit distinct responses to AR SEB impacts, particularly AR-related SWN impacts. Further research is*

*warranted to comprehensively investigate the relationship between Arctic ARs and Arctic cyclones, and their synergistic role in surface SEB impacts, with a particularly focus of cyclone-induced snow on ice. Additionally, it is crucial to compare these findings with the results obtained from ARs in this study."*

*Line 79. There are other mechanisms for extremes (which have a disproportionate impact) of the energy budget eg.*

*Papritz, L., S. Murto, M. Röthlisberger, R. Caballero, G. Messori, G. Svensson, and H. Wernli, 2023: The Role of Local and Remote Processes for Wintertime Surface Energy Budget Extremes over Arctic Sea Ice. J. Climate, 36, 7657–7674, https://doi.org/10.1175/JCLI-D-22-0883.1.*

*But it may be sensible not to extend the length of the submission by avoiding discussion of extremes as this is whole topic in itself.*
Reply: We have incorporated a brief discussion of the mechanisms underlying the SEB events and cited the recommended paper in the manuscript. This addition is as follows:

*"ARs are not solely responsible for the occurrence of extremely large SEB anomalies events, which also involve Arctic air mass and their local transformation (Murto et al., 2023; Papritz et al., 2023). However, gaining a comprehensive understanding of the intricate relationship between ARs and the surface energy budgets provides valuable insights into the remote mechanisms driving Arctic warming, sea ice melt, and changes in the regional climate."*

*Line 95. You should note here that ECMWF does not directly assimilate tropospheric water vapour over land or sea ice, except for radio occultation which does not have the capability to detect AR, and so there is no actual measurements*
Reply: We have included this note in the new Section 6.2-Limitations of the reanalysis data (original Section 4.4), as follows:

*"Notably, ECMWF does not directly assimilate tropospheric water vapor over land or over sea, except for radio occultation, resulting in a lack of actual measurements for detecting ARs."*

*Line 96. If you just did explosive cyclone tracking, would you get the same answer? After all, it is the clouds that matter for LW-down rather than the water vapour itself.*
Reply: We have incorporated this point into the Section 6.1 (original Section 4.3) of our manuscript. But, as we noted above, the research community does consider assessing the impact of ARs as stand-alone features to be an appropriate topic and thus we retain this focus in our manuscript.

*Line 175. Rewrite such that Figure 1 is not the subject of the sentence but supports the statements e.g. 'The seasonal frequency of AR occurrence (Fig 1) shows…*
Reply: We rewrote this sentence as "The spatial distributions of 40-year average AR occurrence frequency (Fig. 1) exhibits prominent seasonality and regional characteristics."

*Line 176. Avoid putting detail in the text which should be in the figure caption (eg. The index used and the limitation of the period 1980-2019. Otherwise, you are repeating what should have been in the methods section. Have a new sentence to introduce the topic of Table 1*
Reply: We have deleted the statement of "1980-2019" and the AR index, and the new sentence was stated above. Because the methods to calculate the Table 1 is detailed in Section 2.3, we only briefly introduce the topic of Table 1 here, as follows:

*"Table 1 summarizes the area averaged AR occurrence frequency for four sub-regions during each season".*

*Table 1. I do not understand this table. The AR are already included in the seasonal climatology so how can they contribute more than 100% of the LWD or surface energy budget? E.g. Greenland. The only way to do this properly is to total the number of J/m2/s for the time without AR and then sum over the time with AR.*
Reply: We have now included the equations used to calculate the metrics evaluating AR's contribution to the net SEB, along with a detailed description of the calculation process for the results presented in Table 1, in Section 2.3. We hope this addition will provide the Reviewer and others reading our manuscript with a clearer understanding of the methodology used for this metric.

References:

Bao, J. W., Michelson, S. A., Neiman, P. J., Ralph, F. M., and Wilczak, J. M.: Interpretation of enhanced integrated water vapor bands associated with extratropical cyclones: Their formation and connection to tropical moisture, Mon Weather Rev, 134, https://doi.org/10.1175/MWR3123.1, 2006.
Blanchard-Wrigglesworth, E., Webster, M., Boisvert, L., Parker, C., and Horvat, C.: Record Arctic Cyclone of January 2022: Characteristics, Impacts, and Predictability, Journal of Geophysical Research: Atmospheres, 127, https://doi.org/10.1029/2022JD037161, 2022.
Dacre, H. F., Clark, P. A., Martinez-Alvarado, O., Stringer, M. A., and Lavers, D. A.: How do atmospheric rivers form?, Bull Am Meteorol Soc, 96, https://doi.org/10.1175/BAMS-D-14-00031.1, 2015.
Eiras-Barca, J., Ramos, A. M., Pinto, J. G., Trigo, R. M., Liberato, M. L. R., and Miguez-Macho, G.: The concurrence of atmospheric rivers and explosive cyclogenesis in the North

Atlantic and North Pacific basins, Earth System Dynamics, 9, https://doi.org/10.5194/esd-9-91-2018, 2018.

Fearon, M. G., Doyle, J. D., Ryglicki, D. R., Finocchio, P. M., and Sprenger, M.: The Role of Cyclones in Moisture Transport into the Arctic, https://doi.org/10.1029/2020GL090353, 2021.

Guo, Y., Shinoda, T., Guan, B., Waliser, D. E., and Chang, E. K. M.: Statistical relationship between atmospheric rivers and extratropical cyclones and anticyclones, J Clim, 33, https://doi.org/10.1175/JCLI-D-19-0126.1, 2020.

Hegyi, B. M. and Taylor, P. C.: The unprecedented 2016–2017 Arctic sea ice growth season: the crucial role of atmospheric rivers and longwave fluxes, Geophys Res Lett, 45, 5204–5212, 2018.

Mattingly, K. S., Mote, T. L., and Fettweis, X.: Atmospheric River Impacts on Greenland Ice Sheet Surface Mass Balance, Journal of Geophysical Research: Atmospheres, 123, https://doi.org/10.1029/2018JD028714, 2018.

Mattingly, K. S., Mote, T. L., Fettweis, X., As, D. V. A. N., Tricht, K. V. A. N., Lhermitte, S., Pettersen, C., and Fausto, R. S.: Strong summer atmospheric rivers trigger Greenland ice sheet melt through spatially varying surface energy balance and cloud regimes, J Clim, 33, https://doi.org/10.1175/JCLI-D-19-0835.1, 2020.

Mattingly, K. S., Turton, J. V., Wille, J. D., Noël, B., Fettweis, X., Rennermalm, Å. K., and Mote, T. L.: Increasing extreme melt in northeast Greenland linked to foehn winds and atmospheric rivers, Nat Commun, 14, https://doi.org/10.1038/s41467-023-37434-8, 2023.

Murto, S., Papritz, L., Messori, G., Caballero, R., Svensson, G., and Wernli, H.: Extreme Surface Energy Budget Anomalies in the High Arctic in Winter, J Clim, 36, https://doi.org/10.1175/JCLI-D-22-0209.1, 2023.

Neiman, P. J., Ralph, F. M., Wick, G. A., Lundquist, J. D., and Dettinger, M. D.: Meteorological characteristics and overland precipitation impacts of atmospheric rivers affecting the West coast of North America based on eight years of SSM/I satellite observations, J Hydrometeorol, 9, https://doi.org/10.1175/2007JHM855.1, 2008.

Papritz, L., Murto, S., Röthlisberger, M., Caballero, R., Messori, G., Svensson, G., and Wernli, H.: The Role of Local and Remote Processes for Wintertime Surface Energy Budget Extremes over Arctic Sea Ice, J Clim, 36, https://doi.org/10.1175/JCLI-D-22-0883.1, 2023.

Ralph, F. M., Neiman, P. J., and Wick, G. A.: Satellite and CALJET aircraft observations of atmospheric rivers over the Eastern North Pacific Ocean during the winter of 1997/98, Mon Weather Rev, 132, https://doi.org/10.1175/1520-0493(2004)132<1721:SACAOO>2.0.CO;2, 2004.

Ralph, F. M., Neiman, P. J., Wick, G. A., Gutman, S. I., Dettinger, M. D., Cayan, D. R., and White, A. B.: Flooding on California's Russian River: Role of atmospheric rivers, Geophys Res Lett, 33, https://doi.org/10.1029/2006GL026689, 2006.

Ralph, F. M., Dettinger, M. C. L. D., Cairns, M. M., Galarneau, T. J., and Eylander, J.: Defining "Atmospheric river" : How the glossary of meteorology helped resolve a debate, Bull Am Meteorol Soc, 99, https://doi.org/10.1175/BAMS-D-17-0157.1, 2018.

Villamil-Otero, G. A., Zhang, J., He, J., and Zhang, X.: Role of extratropical cyclones in the recently observed increase in poleward moisture transport into the Arctic Ocean, Adv Atmos Sci, 35, https://doi.org/10.1007/s00376-017-7116-0, 2018.

Webster, M. A., Parker, C., Boisvert, L., and Kwok, R.: The role of cyclone activity in snow accumulation on Arctic sea ice, Nat Commun, 10, https://doi.org/10.1038/s41467-019-13299-8, 2019.

Zhang, P., Chen, G., Ting, M., Ruby Leung, L., Guan, B., and Li, L.: More frequent atmospheric rivers slow the seasonal recovery of Arctic sea ice, Nat Clim Chang, 13, 266–273, 2023.

Zhang, Z., Ralph, F. M., and Zheng, M.: The Relationship Between Extratropical Cyclone Strength and Atmospheric River Intensity and Position, Geophys Res Lett, 46, https://doi.org/10.1029/2018GL079071, 2019.

---

## Author Comment (AC2)

**egusphere-2024-320**
**"Quantifying the Impacts of Atmospheric Rivers on the Surface Energy Budget of the Arctic Based on Reanalysis"**
**Response to the Reviewers**

**By Chen Zhang, John J. Cassano, Mark Seefeldt, Hailong Wang, Weiming Ma, and Wen-wen Tung**

We appreciate the valuable comments provided by the Reviewers. Before addressing each point individually, we would like to acknowledge the two common concerns raised by Reviewers.

Firstly, there were concerns regarding the methodology of our analysis. The primary objective of this work is to estimate the relative contribution of different surface energy budget (SEB) components to the net SEB. To achieve this, original panel (c) in the Figures 2-3, 5-7 of the manuscript aims to illustrate the relative AR contribution to SEB components, normalized by the net SEB. This normalization involves calculating the ratio of the accumulated AR SEB term, which accounts for both the magnitude of individual AR anomalies and their frequency of occurrence, to the accumulated seasonal net SEB. By adopting this normalization approach, we enable consistent comparisons across different SEB components, thereby allowing readers to discern relative contributions effectively.

Furthermore, following RC3's suggestion with a slight modification, we chose to calculate the relative contribution of AR-related SEB component anomaly normalized by the mean of each respective component. This approach aims to estimate the accumulated AR contribution of SEB component relative to their total values. We chose to present the results as an additional panel, now labeled as new panel (c), in Figures 2-3, and 5-7 of the revised manuscript. Consequently, the original panel (c), depicting the AR SEB contribution normalized by the total SEB, has been reassigned to panel (d) to accommodate this adjustment.

Specifically, the results shown in new panel (c) result from the following calculation at each individual grid point within the study domain for each season:

1. Calculate the total extra energy contributed by each SEB component when ARs are present as, $(F_{AR} - F_{All}) * t_{AR}$, where $F_{AR}$ represents the mean of any term in the SEB equation when an AR is present, $F_{All}$ denotes the seasonal mean of any term in the SEB equation, and $t_{AR}$ indicates the total number of 3-hourly time steps during which ARs are present.

2. Calculate the total energy for each component as, $F_{All} * t_{All}$, where $t_{All}$ signifies the total number of 3-hourly time steps within each season.

3. Determine the ratio of these two terms, which provides an estimate of the magnitude of AR anomaly for each SEB term relative to the average value for each component. This is

presented in Eq. (2), noting that the ratio of $t_{AR}$ to $t_{All}$ is simply the AR frequency shown in Fig. 1

$$\frac{(F_{AR} - F_{All}) * t_{AR}}{F_{All} * t_{All}} = \frac{panel\ (b) * Fig.1}{panel\ (a)} \tag{2}$$

Additionally, we include the net SEB equation in the revised manuscript, labeled as Eq. (1), as follows:

$$net\ SEB = LWN + SWN + TH = LWD - LWU + SWD - SWU + SH + LH \tag{1}$$

Where LWN, SWN and TH denote the net longwave radiation, net shortwave radiation, and turbulent heat flux, respectively. LWD, LWU, SWD, SWU, SH, LH represent downward longwave, upward longwave, downward shortwave, upward shortwave, sensible and latent heat flux, respectively.

Secondly, two Reviewers expressed concerns about the organization of our sections, particularly noting overlapping discussions between Section 3 (Analysis and Results) and Section 4 (Discussion). To address this issue, we have restructured the sections as follows:

- Section 3: AR occurrence frequency (original Section 3.1)
- Section 4: AR's influence on the surface energy budget component of the Arctic (original Section 3.2)
    - Section 4.1: Surface radiative fluxes (original Section 3.2.1)
        - Section 4.1.1: Surface downward longwave radiation
        - Section 4.1.2: Net Surface longwave radiation
        - Section 4.1.3: Net Surface shortwave radiation
    - Section 4.2: Surface turbulent heat fluxes (original Section 3.2.2)
    - Section 4.3: Net Surface energy budget (original Section 3.2.3)
- Section 5: AR's surface impacts
    - Section 5.1: AR-induced surface and air temperature response (original Section 4.1)
    - Section 5.2: AR's crucial role in triggering Greenland Ice Sheet melt (original Section 4.2)
- Section 6: Uncertainties and limitations
    - Section 6.1: Influence of AR detection methods on results (original Section 4.3)
    - Section 6.2: Limitation of the reanalysis data (original Section 4.4)
- Section 7: Conclusions (original Section 5)

We believe these adjustments will enhance the clarity and coherence of our manuscript, addressing the concerns raised by the Reviewers effectively.

Below, we respond in blue text to the Reviewer's comments, using an italic font to indicate text that has been copied verbatim from the Reviewer's reports.
* * *
Reply to RC2, Jonathan Wille:
We appreciate the Reviewer for insightful and detailed reviews. We have made changes to the manuscript, accordingly, as replied below.

***RC2, Jonathan Wille:***
***General comments***

*This study is a comprehensive examination of the atmospheric river (AR) influence of the surface energy budget (SEB) across the entire Arctic. Using an AR detection algorithm based on relative monthly integrated vapor transport (IVT), the authors identify the distinctions of AR SEB influence across land, open ocean, and sea ice regions. Their results help confirm and build upon previous understandings about AR impacts on Greenland surface melting and especially the hampering of winter sea-ice growth. Regarding this AR impact on winter sea-ice growth, the observation that this process is highly sensitive to the choice of AR detection algorithm is a great distinction between the impacts observed while using an AR detection algorithm designed to capture extreme events and an algorithm designed to capture more frequent events. There is a clear line of progression from the authors' previous first work on Arctic AR climatology to this study on Arctic AR SEB behavior. The methods are clear and well formulated, and the results are exhaustive and detailed. To my knowledge, previous studies have looked at localized Arctic SEB impacts from ARs, but this is the first study to make a comprehensive analysis on this topic across the entire Arctic. After some minor revisions, this manuscript will serve as an excellent reference for other researchers looking to understand the overall influence of ARs on the polar SEB. I would be happy to see this manuscript published after some global comments and a series of minor comments are addressed.*

Reply: Thank you for the positive comments.

*Specific comments*

1. *Section 2.3: Please consider including the equation for the SEB so the reader can quickly understand the various SEB components presented in this manuscript.*

Reply: We have included the equation for the SEB in Section 2.2, as Eq. (1) in the manuscript:

*"Moreover, we define total surface turbulent heat flux (TH) as the sum of SH and LH. The net SEB is expressed as the sum of the net radiation at the surface (i.e., sum of the LWN and SWN) and net total TH (i.e., sum of the SH and LH), that is,*
$$net\ SEB = LWN + SWN + TH = LWD - LWU + SWD - SWU + SH + LH \qquad (1)$$
*Where LWU and SWU represent upward longwave and shortwave radiation, respectively."*

2. *Sec 3.1: Please discuss how the AR frequency results presented here compare to the analysis in Zhang et al., (2023). Assuming that this is a similar analysis as Zhang et al., (2023), it may be helpful to mention that you have repeated this AR frequency analysis to*

*help contextualize your later SEB results. I do like that you made this a small section as to not detract from the SEB analysis.*

Reply: It is acknowledged that Sec 3.1 presents a similar analysis compared to that conducted in Zhang et al., (2023). In response, we have included a clarifying statement in the manuscript, stating:

*"It is noted that the AR occurrence frequency presented in Fig. 1 resembles the analysis in Zhang et al., (2023), with the distinction that we emphasize the seasonal frequency as a percentage of total time steps within each season instead of annual percentage."*

3. *Figure order: Consider changing the order of the results so that the net SEB is presented first and followed by the components of the SEB. This could improve the readability since currently Figure 7 is referenced before Figures 3-6 when discussing the LWD results.*

Reply: We appreciate the Reviewer's suggestion, but we prefer to retain the order of figures and discussion as originally shown in our manuscript. Our rationale for this is that many previous Arctic AR studies highlight the large impact of ARs on longwave radiation and thus we chose to begin our discussion with this SEB term. We then feel that it makes sense to proceed through other individual terms in the SEB and ending with the net SEB, which sums the previously discussed results.

4. *Section 4.2: This is a good discussion comparing the melting implications of your study with previous works, but it could use some more elaboration and clarity. In the beginning, you mention is disparity between the results of Mattingly et al., 2020 which found ARs delivered large sensible heat fluxes while your study links ARs to smaller turbulent heat fluxes and more net longwave anomalies. You attribute these differences to the focus on stronger ARs in Mattingly et al., 2020, but could elaborate on why a focus on stronger ARs might cause these differences?*

Reply: The discrepancy from Mattingly et al., (2020) could be attributed to the use of distinct AR detection algorithms. Their approach applies a stringent minimum threshold of 150 kg m$^{-1}$ s$^{-1}$ for IVT and exclusively allows for northward moisture transport from the Arctic, potentially leading to highly intense northward AR transport and heightened sensible heat flux. We have elaborated on this discussion as follows:

*"This deviation from Mattingly et al., (2020) may stem from the utilization of different AR detection algorithms. Their methodology imposes a strict minimum threshold of 150 kg m$^{-1}$ s$^{-1}$ for IVT, and exclusively considers northward moisture transport from the Arctic. Moreover, their focus is on the strongest AR days, where the maximum IVT exceeds the 90th percentile of all AR IVT at each basin and each season. These criteria are designed to capture extremely strongly northward AR transport events affecting Greenland, potentially resulting in heightened SH."*

*Then you discuss your findings in Northeast Greenland which point to a larger influence of turbulent heat fluxes which actually agrees a bit with Mattingly et al., 2020 and aligns closer to Mattingly et al., (2023) which discusses more the foehn effect from ARs. It would be good if you can mention this agreement with Mattingly et al., (2023) and how your sensible heat flux results might be picking up on the AR-related Foehn contribution in the region.*

Reply: We have integrated this discussion into the manuscript as follows:

*"These patterns align with findings from Mattingly et al., (2023, 2020), where they suggest that the foehn effect from ARs leads to increased SH."*

5. *Section 4.3: Naturally, some readers will wonder if you would get similar results using a cyclone-detection algorithm to study SEB impacts. I'm not suggesting you make an additional analysis with a cyclone-detection algorithm, but perhaps it could be beneficial to add a few sentences to the end of this section relating your results with other studies that did track SEB-impacts from cyclones and then argue why it is more informative to use ARs instead of cyclones to quantify SEB-impacts.*

Reply: ARs are indeed strongly associated with cyclones. Exploring the role of Arctic cyclones in SEB impacts and comparing them with results of ARs in this study is a direction that requires further investigation. We have incorporated this point into the Section 6.1 (original Section 4.3) of our manuscript as follows:

*"In addition, Arctic ARs are closely linked with Arctic cyclones, which strongly influence surface heat fluxes, particularly TH (Blanchard-Wrigglesworth et al., 2022), subsequently impacting the net SEB. Moreover, studies suggest that large SEB anomaly events in the Arctic are often associated with an increased frequency of cyclone occurrence (Murto et al., 2023). Additionally, cyclones affect snowfall accumulation on sea ice, thereby influencing SEB due to high albedo of snow (Webster et al., 2019). Our findings indicate that surfaces with varying albedos exhibit distinct responses to AR SEB impacts, particularly AR-related SWN impacts. Further research is warranted to comprehensively investigate the relationship between Arctic ARs and Arctic cyclones, and their synergistic role in surface SEB impacts, with a particularly focus of cyclone-induced snow on ice. Additionally, it is crucial to compare these findings with the results obtained from ARs in this study."*

*Minor comments*

*Line 29: First sentence is a run-on. Consider breaking it up.*

Reply: We have broken the sentence into two sentences, as follows:

*"The Arctic is a multifaceted environment, distinguished by close interactions among its atmosphere, ocean, sea ice and land components. It is influenced by various forcing from lower latitudes, operating across a wide range of time and space scales (Serreze et al., 2007)."*

*Line 41: "Remote perspective" is slightly vague. Maybe "remote forcing perspective"*
Reply: We have changed to "remote forcing perspective". Thank you.

*Line 47: Consider distinguishing the studies that focus on Antarctic ARs and Arctic ARs.*
Reply: We have categorized the literature into Arctic and Antarctic ARs, and expanded our references on Antarctic ARs, as follows:

*"This growing attention is evident in various Arctic studies (Baggett et al., 2016; Ma et al., 2021; Mattingly et al., 2023, 2020; Zhang et al., 2023a, b) and Antarctic studies (Gorodetskaya et al., 2014; Guan et al., 2016; Ma et al., 2020; Wille et al., 2021; Shields et al., 2022; Wille et al., 2019, 2024b, a)…"*

*Line 53: Add an oxford comma after "ocean"*
Reply: Added.

*Line 67-68: It's good you cited the importance of the AR impacts on the SEB in relation to sea ice. But since this paper also discusses the SEB over land, you should also state the importance of the AR SEB impacts over land ice.*
Reply: We have incorporated the importance of AR SEB impacts over land ice, as follows:

*"… Moreover, the impacts of AR on the SEB can extend beyond sea ice regions to encompass land ice dynamics. These impacts include various facets, including melting rates, warming of the snowpack, affecting snowmelt timing, alterations in ice mass balance, and overall surface energy exchange process (Goldenson et al., 2018; Guan et al., 2016)."*

*Line 68: "accelerate or decelerate ice growth" you should clarify that you refer to sea ice growth here.*
Reply: We have changed to "sea ice growth".

*Line 80: Correct "AR's impact" to "AR impacts on the Arctic surface energy budget". Surface energy budget should be singular unless you reference multiple locations in the sentence.*
Reply: We have corrected "AR impacts", and we have also replaced "surface energy budget" with the abbreviation of "SEB" consistently throughout the text.

*Line 86-91: This is a really long sentence. Consider breaking it up around when you describe MERRA-2 being the source data for ARTMIP.*

Reply: We have rewritten this sentence, as follows:

*"An ensemble Arctic AR index database (Tung et al., 2023) was developed by Zhang et al., (2023a), where a total of 12 AR indices were created based on combinatory conditions of either integrated water vapor transport (IVT) or integrated water vapor (IWV) applied with three levels of monthly climate thresholds (75th, 85th, and 95th percentiles). The data utilized for this development were sourced from 3-hourly fifth generation of ECMWF atmospheric reanalysis (ERA5, Hersbach et al., 2020) and 3-hourly NASA Modern-Era Retrospective Analysis for Research and Applications, version 2 (MERRA-2, Gelaro et al., 2017) from 1980 to 2019. The NASA MERRA-2 source data was obtained from the AR Tracking Method Intercomparison Project (Shields et al., 2018)."*

*Line 100: Can you briefly say why you only choose dates during neutral or weak ENSO events?*
Reply: We only preserved the dates during neutral or weak ENSO events to have a standard climate threshold to test for ARs. For example, if we wanted to update the AR index to the MOSAiC year, we do not need to collect the data and recalculate the thresholds. To clarify this approach, we have included the following note in the manuscript:

*"The selection of the neutral or weak ENSO events aim to establish a standard climate threshold for testing ARs."*

*Line 133: It seems odd that surface energy budget is first abbreviated here and not earlier in the introduction on its first use.*
Reply: We have addressed the issue by removing the abbreviation in this instance and ensuring consistency in the use of "SEB" throughout the manuscript.

*Line 159: Consider changing to "underscores the potential role of ARs driving net SEB fluctuations"*
Reply: Changed it.

*Line 164: Comma after "To do this"*
Reply: Fixed.

*Section 3.2: Just wanted to say that I appreciate you outlining the different figures and tables here before continuing to the sub-sections. This is very helpful for the reader to follow along.*
Reply: Thank you for this comment.

*Line 227-228: Nice result here. You could comment that ARs are nearly the exclusive cause of LWD over Arctic land areas during winter. This would make them the main cause for warming during the winter since winter warming is driven by LWD*

Reply: We are cautious about drawing this conclusion solely based on the results presented in the manuscript. As stated at the beginning, we will calculate the relative contribution of AR-related SEB components normalized by the mean of each respective component. These findings will be presented in the new panel (c) in the revised manuscript, providing further insights to verify the suggested comment.

*Figure 2c,3c,5c,6c,7c: On both ends of the color bar, there is a gray color to represent values exceeding -100 and 100%. In Figure 2c, the caption says these gray areas represent percentage results greater than 100%. However, in some other figures, the gray areas represent percentages less than 100%, but this isn't mentioned in their figure captions. Please clarify this either in the Figure 2 caption or the following figure captions.*

Reply: We consistently use the gray color to represent the percentage results greater than 100% or less than -100% across Figs 2-3c,5-7c. We have adjusted the description to accurately reflect this:

*"The percentage results greater than 100% or less than -100% are shaded in grey for clarity."*

*Line 257: Figure 7 is cited before Figures 3-6. While I appreciate that this is meant to enhance the discussion of the results in Figure 2, it is disorientating to the reader since they haven't had a chance to understand the meaning of Figure 7 and forces them to skip ahead in the manuscript. Please considering moving Figure 7 to Figure 3, moving this text to the discussion, or devise another solution to improve the order of results here.*

Reply: Please see our response to your specific comment 3.

*Line 274: You mean the AR-related LWD contribution here?*

Reply: Yes, we have changed it to "AR-related LWD contribution".

*Line 276-284: While I do like some reflection on the meaning of the results in the Results section, this paragraph feels more appropriate for the Discussion section. Especially since you are citing Figure 7 before Figures 3-6.*

Reply: In addressing this concern, we have retitled this section to specifically focus on surface downward longwave radiation.

*Line 294: This might be a question for the editor, but it would be helpful if there was some label or subsection break between the LWD and LWN results (and for the other SEB components). Even just "Net surface long radiation" written in bold would help the reader follow along.*

Reply: We have reorganized the manuscript. This part has been rearranged as a new subsection, titled "Section 4.1.2 Net surface longwave radiation".

*Line 328: You should cite Zhang et al., (2023b) here concerning the AR impacts on marginal sea ice zones*
Reply: We have included this citation.

*Figure 3: Is there a particular reason why it appears AR have a negative LWN contribution in this patch over central Siberia?*
Reply: We are also surprised by the negative contribution of ARs over central Siberia in Figure 3, but we do not have any thoughts as to why this occurs. We do note that the negative anomaly values in this region are quite small, ranging from -0 to -4.5 W m$^{-2}$.

*Line 360-361: Does this mean that AR-related warm air advection is more important than the AR-related SEB influence?*
Reply: There's no necessity for this to imply that AR-related warm air advection is more important than the AR-related SEB influence. Our observation simply indicates that the AR-related surface temperature response is more closely associated with the AR-related LWD effects compared to AR-related SEB impacts.

*Line 394: Add space in (Fig.5b).*
Reply: Corrected.

*Line 408-409: Here and other places you should clarify that this warming role is confined to the SEB and does not include warm air advection related to ARs.*
Reply: We explicitly mentioned that "AR-related LWN and SWN anomalies differ by less than 1 W m$^{-2}$, indicating little overall radiative impact of ARs at this time of the year over ice-covered surfaces…". Therefore, the assertion regarding warming is specific to the radiative perspective associated with ARs, including the longwave and shortwave radiation that is being analyzed in the current section of the manuscript. It does not encompass the sensible heat flux associated with warm air advection related to ARs.

*Line 414: Play not plays*
Reply: Fixed.

*Line 415: Add oxford comma.*
Reply: Added.

*Line 441: Comma after "Unlike other Arctic regions"*
Reply: Added.

*Line 456: Comma after "Arctic Ocean"*
Reply: Done.

*Line 474: Add "the" between "highlights AR's"*
Reply: Added.

*Line 474-483: I was wondering why the AR contribution to turbulent heat fluxes is negative around the coastline of Greenland, but positive over the Greenland interior. Perhaps you can comment on this in this last paragraph of the section.*
Reply: The anomaly values (Fig. 6b) around the coastline of Greenland are indeed very small in terms of magnitude, ranging from -11 $m^{-2}$ to -0. These values may be influenced by the complex geographic features present in high latitudes near the coastline. Additionally, we observe distinct differences in AR-related turbulent heat flux patterns between Greenland interior and the surrounding ocean areas. The coastlines of Greenland serve as a transition zone between the land and ocean, resulting in complicated TH features. However, the specific reasons for the negative anomalies observed around the coastlines require further investigation. We have included a comment at the end of this paragraph to acknowledge the need for additional research:

*"Additionally, AR-related TH features over the coastlines are different from those observed over the Greenland interior, such as the presence of weak negative anomalies and corresponding negative AR contribution to the net SEB. Further exploration is necessary to understand the underlying factors contributing to these features."*

*Line 506: This delay in sea-ice refreezing is also a result from Zhang et al., (2023b) and should be mentioned here.*
Reply: We have added this reference.

*Line 529: Comma after "central Arctic"*
Reply: Added.

*Line 559-561: I'm very happy to see these AR temperature anomalies quantified so extensively for the Arctic region*
Reply: Thank you for this comment.

*Line 657-659: This remark about the sensitivity of the AR effect on the hampering of the winter sea-ice freeze to the choice in detection method is one of the more compelling implications from this study. You make a great point about the risks of only capturing extreme AR events for studying impacts. Although not necessary, this would be a good point to include in the abstract if you can replace another sentence as the abstract is already long.*
Reply: We agree that this is one of the important results in this paper, which highlights that different AR detection methods may lead to different physical results. Therefore, we have included this point in the abstract, as follows:

*"... Additionally, AR-related SEB impacts strongly rely on AR detection methods, as the use of restrictive AR detection algorithms tends to emphasize extreme AR events but may minimize their total contribution to the SEB climatology due to their low occurrence frequency. Overall, this study quantifies the role of ARs on surface energy budget, contribution to our understanding of the Arctic warming and sea ice decline in ongoing Arctic amplification."*

*Line 675: "rely" not "relies"*
Reply: Fixed.

*Line 693-695: Suggest rewording this sentence. "partially offsetting the large LWD anomalies, thus resulting in moderate impacts on the LWN anomalies"*
Reply: We have adjusted this sentence accordingly. Thank you!

*Line 727-728: "especially during cold seasons, particularly winter". Suggest rephrasing since most people would consider winter the cold season.*
Reply: We have removed "especially".

*References:*

*Mattingly, K. S., Turton, J. V., Wille, J. D., Noël, B., Fettweis, X., Rennermalm, Å. K., and Mote, T. L.: Increasing extreme melt in northeast Greenland linked to foehn winds and atmospheric rivers, Nature Communications, 14, 1743, https://doi.org/10.1038/s41467-023-37434-8, 2023.*

*Zhang, C., Tung, W., and Cleveland, W. S.: Climatology and decadal changes of Arctic atmospheric rivers based on ERA5 and MERRA-2, Environ. Res.: Climate, 2, 035005, https://doi.org/10.1088/2752-5295/acdf0f, 2023a.*

*Zhang, P., Chen, G., Ting, M., Ruby Leung, L., Guan, B., and Li, L.: More frequent atmospheric rivers slow the seasonal recovery of Arctic sea ice, Nature Climate Change, 13, 266–273, https://doi.org/10.1038/s41558-023-01599-3, 2023b.*

References
    Baggett, C., Lee, S., and Feldstein, S.: An investigation of the presence of atmospheric rivers over the North Pacific during planetary-scale wave life cycles and their role in Arctic warming, J Atmos Sci, 73, https://doi.org/10.1175/JAS-D-16-0033.1, 2016.
    Blanchard-Wrigglesworth, E., Webster, M., Boisvert, L., Parker, C., and Horvat, C.: Record Arctic Cyclone of January 2022: Characteristics, Impacts, and Predictability, Journal of Geophysical Research: Atmospheres, 127, https://doi.org/10.1029/2022JD037161, 2022.

Gelaro, R., McCarty, W., Suárez, M. J., Todling, R., Molod, A., Takacs, L., Randles, C. A., Darmenov, A., Bosilovich, M. G., Reichle, R., Wargan, K., Coy, L., Cullather, R., Draper, C., Akella, S., Buchard, V., Conaty, A., da Silva, A. M., Gu, W., Kim, G. K., Koster, R., Lucchesi, R., Merkova, D., Nielsen, J. E., Partyka, G., Pawson, S., Putman, W., Rienecker, M., Schubert, S. D., Sienkiewicz, M., and Zhao, B.: The modern-era retrospective analysis for research and applications, version 2 (MERRA-2), J Clim, 30, https://doi.org/10.1175/JCLI-D-16-0758.1, 2017.

Goldenson, N., Leung, L. R., Bitz, C. M., and Blanchard-Wrigglesworth, E.: Influence of atmospheric rivers on mountain snowpack in the western United States, J Clim, 31, https://doi.org/10.1175/JCLI-D-18-0268.1, 2018.

Gorodetskaya, I. V., Tsukernik, M., Claes, K., Ralph, M. F., Neff, W. D., and Van Lipzig, N. P. M.: The role of atmospheric rivers in anomalous snow accumulation in East Antarctica, Geophys Res Lett, 41, https://doi.org/10.1002/2014GL060881, 2014.

Guan, B., Waliser, D. E., Ralph, F. M., Fetzer, E. J., and Neiman, P. J.: Hydrometeorological characteristics of rain-on-snow events associated with atmospheric rivers, Geophys Res Lett, 43, https://doi.org/10.1002/2016GL067978, 2016.

Hersbach, H., Bell, B., Berrisford, P., Hirahara, S., Horányi, A., Muñoz-Sabater, J., Nicolas, J., Peubey, C., Radu, R., Schepers, D., Simmons, A., Soci, C., Abdalla, S., Abellan, X., Balsamo, G., Bechtold, P., Biavati, G., Bidlot, J., Bonavita, M., De Chiara, G., Dahlgren, P., Dee, D., Diamantakis, M., Dragani, R., Flemming, J., Forbes, R., Fuentes, M., Geer, A., Haimberger, L., Healy, S., Hogan, R. J., Hólm, E., Janisková, M., Keeley, S., Laloyaux, P., Lopez, P., Lupu, C., Radnoti, G., de Rosnay, P., Rozum, I., Vamborg, F., Villaume, S., and Thépaut, J. N.: The ERA5 global reanalysis, Quarterly Journal of the Royal Meteorological Society, 146, https://doi.org/10.1002/qj.3803, 2020.

Ma, W., Chen, G., and Guan, B.: Poleward Shift of Atmospheric Rivers in the Southern Hemisphere in Recent Decades, Geophys Res Lett, 47, https://doi.org/10.1029/2020GL089934, 2020.

Ma, W., Chen, G., Peings, Y., and Alviz, N.: Atmospheric River Response to Arctic Sea Ice Loss in the Polar Amplification Model Intercomparison Project, Geophys Res Lett, 48, https://doi.org/10.1029/2021GL094883, 2021.

Mattingly, K. S., Mote, T. L., Fettweis, X., As, D. V. A. N., Tricht, K. V. A. N., Lhermitte, S., Pettersen, C., and Fausto, R. S.: Strong summer atmospheric rivers trigger Greenland ice sheet melt through spatially varying surface energy balance and cloud regimes, J Clim, 33, https://doi.org/10.1175/JCLI-D-19-0835.1, 2020.

Mattingly, K. S., Turton, J. V., Wille, J. D., Noël, B., Fettweis, X., Rennermalm, Å. K., and Mote, T. L.: Increasing extreme melt in northeast Greenland linked to foehn winds and atmospheric rivers, Nat Commun, 14, https://doi.org/10.1038/s41467-023-37434-8, 2023.

Murto, S., Papritz, L., Messori, G., Caballero, R., Svensson, G., and Wernli, H.: Extreme Surface Energy Budget Anomalies in the High Arctic in Winter, J Clim, 36, https://doi.org/10.1175/JCLI-D-22-0209.1, 2023.

Serreze, M. C., Barrett, A. P., Slater, A. G., Steele, M., Zhang, J., and Trenberth, K. E.: The large-scale energy budget of the Arctic, Journal of Geophysical Research: Atmospheres, 112, 2007.

Shields, C. A., Rutz, J. J., Leung, L. Y., Martin Ralph, F., Wehner, M., Kawzenuk, B., Lora, J. M., McClenny, E., Osborne, T., Payne, A. E., Ullrich, P., Gershunov, A., Goldenson, N., Guan, B., Qian, Y., Ramos, A. M., Sarangi, C., Sellars, S., Gorodetskaya, I., Kashinath, K., Kurlin, V., Mahoney, K., Muszynski, G., Pierce, R., Subramanian, A. C., Tome, R., Waliser, D., Walton, D., Wick, G., Wilson, A., Lavers, D., Prabhat, Collow, A., Krishnan, H., Magnusdottir, G., and Nguyen, P.: Atmospheric River Tracking Method Intercomparison Project (ARTMIP): Project goals and experimental design, Geosci Model Dev, 11, https://doi.org/10.5194/gmd-11-2455-2018, 2018.

Shields, C. A., Wille, J. D., Marquardt Collow, A. B., Maclennan, M., and Gorodetskaya, I. V.: Evaluating Uncertainty and Modes of Variability for Antarctic Atmospheric Rivers, Geophys Res Lett, 49, https://doi.org/10.1029/2022GL099577, 2022.

Tung, W., Zhang, C., and Cleveland, W. S.: Arctic Atmospheric River Labels and Climatology Based on 3-hourly ERA5 and MERRA-2 From 1980 to 2019. , Purdue University Research Repository. , 2023.

Webster, M. A., Parker, C., Boisvert, L., and Kwok, R.: The role of cyclone activity in snow accumulation on Arctic sea ice, Nat Commun, 10, https://doi.org/10.1038/s41467-019-13299-8, 2019.

Wille, J. D., Favier, V., Dufour, A., Gorodetskaya, I. V., Turner, J., Agosta, C., and Codron, F.: West Antarctic surface melt triggered by atmospheric rivers, Nat Geosci, 12, https://doi.org/10.1038/s41561-019-0460-1, 2019.

Wille, J. D., Favier, V., Gorodetskaya, I. V., Agosta, C., Kittel, C., Beeman, J. C., Jourdain, N. C., Lenaerts, J. T. M., and Codron, F.: Antarctic Atmospheric River Climatology and Precipitation Impacts, Journal of Geophysical Research: Atmospheres, 126, https://doi.org/10.1029/2020JD033788, 2021.

Wille, J. D., Alexander, S. P., Amory, C., Baiman, R., Barthélemy, L., Bergstrom, D. M., Berne, A., Binder, H., Blanchet, J., Bozkurt, D., Bracegirdle, T. J., Casado, M., Choi, T., Clem, K. R., Codron, F., Datta, R., Di Battista, S., Favier, V., Francis, D., Fraser, A. D., Fourré, E., Garreaud, R. D., Genthon, C., Gorodetskaya, I. V., González-Herrero, S., Heinrich, V. J., Hubert, G., Joos, H., Kim, S. J., King, J. C., Kittel, C., Landais, A., Lazzara, M., Leonard, G. H., Lieser, J. L., Maclennan, M., Mikolajczyk, D., Neff, P., Ollivier, I., Picard, G., Pohl, B., Ralph, F. M., Rowe, P., Schlosser, E., Shields, C. A., Smith, I. J., Sprenger, M., Trusel, L., Udy, D., Vance, T., Vignon, É., Walker, C., Wever, N., and Zou, X.: The Extraordinary March 2022 East Antarctica "Heat" Wave. Part I: Observations and Meteorological Drivers, J Clim, 37, https://doi.org/10.1175/JCLI-D-23-0175.1, 2024a.

Wille, J. D., Alexander, S. P., Amory, C., Baiman, R., Barthélemy, L., Bergstrom, D. M., Berne, A., Binder, H., Blanchet, J., Bozkurt, D., Bracegirdle, T. J., Casado, M., Choi, T., Clem, K. R., Codron, F., Datta, R., Di Battista, S., Favier, V., Francis, D., Fraser, A. D., Fourré, E.,

Garreaud, R. D., Genthon, C., Gorodetskaya, I. V., González-Herrero, S., Heinrich, V. J., Hubert, G., Joos, H., Kim, S. J., King, J. C., Kittel, C., Landais, A., Lazzara, M., Leonard, G. H., Lieser, J. L., Maclennan, M., Mikolajczyk, D., Neff, P., Ollivier, I., Picard, G., Pohl, B., Ralph, F. M., Rowe, P., Schlosser, E., Shields, C. A., Smith, I. J., Sprenger, M., Trusel, L., Udy, D., Vance, T., Vignon, É., Walker, C., Wever, N., and Zou, X.: The Extraordinary March 2022 East Antarctica "Heat" Wave. Part II: Impacts on the Antarctic Ice Sheet, J Clim, 37, https://doi.org/10.1175/JCLI-D-23-0176.1, 2024b.

Zhang, C., Tung, W., and Cleveland, W. S.: Climatology and decadal changes of Arctic atmospheric rivers based on ERA5 and MERRA-2, Environmental Research: Climate, https://doi.org/10.1088/2752-5295/acdf0f, 2023a.

Zhang, P., Chen, G., Ting, M., Ruby Leung, L., Guan, B., and Li, L.: More frequent atmospheric rivers slow the seasonal recovery of Arctic sea ice, Nat Clim Chang, 13, 266–273, 2023b.

---

## Author Comment (AC3)

**egusphere-2024-320 "Quantifying the Impacts of Atmospheric Rivers on the Surface Energy Budget of the Arctic Based on Reanalysis" Response to the Reviewers**

**By Chen Zhang, John J. Cassano, Mark Seefeldt, Hailong Wang, Weiming Ma, and Wenwen Tung**

We appreciate the valuable comments provided by the Reviewers. Before addressing each point individually, we would like to acknowledge the two common concerns raised by Reviewers.

Firstly, there were concerns regarding the methodology of our analysis. The primary objective of this work is to estimate the relative contribution of different surface energy budget (SEB) components to the net SEB. To achieve this, original panel (c) in the Figures 2-3, 5-7 of the manuscript aims to illustrate the relative AR contribution to SEB components, normalized by the net SEB. This normalization involves calculating the ratio of the accumulated AR SEB term, which accounts for both the magnitude of individual AR anomalies and their frequency of occurrence, to the accumulated seasonal net SEB. By adopting this normalization approach, we enable consistent comparisons across different SEB components, thereby allowing readers to discern relative contributions effectively.

Furthermore, following RC3's suggestion with a slight modification, we chose to calculate the relative contribution of AR-related SEB component anomaly normalized by the mean of each respective component. This approach aims to estimate the accumulated AR contribution of SEB component relative to their total values. We chose to present the results as an additional panel, now labeled as new panel (c), in Figures 2-3, and 5-7 of the revised manuscript. Consequently, the original panel (c), depicting the AR SEB contribution normalized by the total SEB, has been reassigned to panel (d) to accommodate this adjustment.

Specifically, the results shown in new panel (c) result from the following calculation at each individual grid point within the study domain for each season:

- 1. Calculate the total extra energy contributed by each SEB component when ARs are present as,  $(F_{AR} F_{All}) * t_{AR}$ , where  $F_{AR}$  represents the mean of any term in the SEB equation when an AR is present,  $F_{All}$  denotes the seasonal mean of any term in the SEB equation, and  $t_{AR}$  indicates the total number of 3-hourly time steps during which ARs are present.
- 2. Calculate the total energy for each component as,  $F_{All} * t_{All}$ , where  $t_{All}$  signifies the total number of 3-hourly time steps within each season.
- 3. Determine the ratio of these two terms, which provides an estimate of the magnitude of AR anomaly for each SEB term relative to the average value for each component. This is

presented in Eq. (2), noting that the ratio of  $t_{AR}$  to  $t_{All}$  is simply the AR frequency shown in Fig. 1

$$\frac{(F_{AR} - F_{All}) * t_{AR}}{F_{All} * t_{All}} = \frac{\text{panel (b)} * \text{Fig.1}}{\text{panel (a)}}$$
(2)

Additionally, we include the net SEB equation in the revised manuscript, labeled as Eq. (1), as follows:

net SEB = LWN + SWN + TH = LWD - LWU + SWD - SWU + SH + LH (1) Where LWN, SWN and TH denote the net longwave radiation, net shortwave radiation, and turbulent heat flux, respectively. LWD, LWU, SWD, SWU, SH, LH represent downward longwave, upward longwave, downward shortwave, upward shortwave, sensible and latent heat flux, respectively.

Secondly, two Reviewers expressed concerns about the organization of our sections, particularly noting overlapping discussions between Section 3 (Analysis and Results) and Section 4 (Discussion). To address this issue, we have restructured the sections as follows:

- Section 3: AR occurrence frequency (original Section 3.1)
- Section 4: AR's influence on the surface energy budget component of the Arctic (original Section 3.2)
  - Section 4.1: Surface radiative fluxes (original Section 3.2.1)
    - Section 4.1.1: Surface downward longwave radiation
    - Section 4.1.2: Net Surface longwave radiation
    - Section 4.1.3: Net Surface shortwave radiation
  - Section 4.2: Surface turbulent heat fluxes (original Section 3.2.2)
  - Section 4.3: Net Surface energy budget (original Section 3.2.3)
- Section 5: AR's surface impacts
  - Section 5.1: AR-induced surface and air temperature response (original Section 4.1)
  - Section 5.2: AR's crucial role in triggering Greenland Ice Sheet melt (original Section 4.2)
- Section 6: Uncertainties and limitations
  - Section 6.1: Influence of AR detection methods on results (original Section 4.3)
  - Section 6.2: Limitation of the reanalysis data (original Section 4.4)
- Section 7: Conclusions (original Section 5)

We believe these adjustments will enhance the clarity and coherence of our manuscript, addressing the concerns raised by the Reviewers effectively.

Below, we respond in blue text to the Reviewer's comments, using an italic font to indicate text that has been copied verbatim from the Reviewer's reports.

\_\_\_\_\_

**Reply to RC3:**

We appreciate the Reviewer for insightful and detailed reviews. We have made changes to the manuscripts, accordingly, as replied below.

**RC3*:**

In this paper, authors aim to analyze the contribution of atmospheric rivers (ARs) to the seasonal surface energy budget (SEB) in the Arctic using ERA5 reanalysis data for 1980-2019. ARs are detected using the 85th percentile of IVT and components of the seasonal SEB are anomalies are assessed for times when ARs are detected. The aim of improving understanding the importance of ARs in net SEB in the Arctic is important and interesting, and the authors provide a very detailed analysis with discussion of implications and connections to previous work. Analysis regarding absolute anomalies is thorough, but I am unsure of the appropriateness of the metric used to quantify the contributions of ARs to seasonal SEB (detailed in Major Comment 1).

Reply: We appreciate the Reviewer's dedication to scrutinizing our metric and proposing new approaches to evaluate the contributions of ARs to net SEB.

**Major Comments:**

1. The metric used for evaluating the contribution of ARs to net SEB may not be appropriate for the conclusions drawn. It is difficult to interpret the physical meaning of the contributions when the seasonal net SEB is very small. Please see the attached file for a description of a potential solution and further reasoning. Regardless of how the authors proceed, the equation used to calculate this metric should be included, rather than only described in words to make sure it is very clear what is being shown.

Reply: We appreciate Reviewer's diligent examination of our original metric, proposing and detailed description of two new metrics we could use to evaluate the contribution of ARs to net SEB. As the Reviewer rightly pointed out, the main goal of this work is to estimate the relative contribution of different SEB components to the net SEB. To achieve this, we utilize original panel (c) in the Figures 2-3, 5-7 of the manuscript to illustrate the relative AR contribution to SEB components, normalized by the net SEB. This normalization involves calculating the ratio of the accumulated AR SEB term, which accounts for both the magnitude of individual AR anomalies and their frequency of occurrence, to the accumulated seasonal net SEB.

A relative contribution exceeding 100% indicates that the considered term has a greater AR contribution than the total SEB, implying that other SEB terms counteract to yield a small net SEB. Very large relative contributions indicate that the climatological SEB results from a small difference in large, oppositely signed terms in the SEB and that one of those large terms has a large AR signal. We believe that this is useful information to show since this normalization facilitates consistent comparison across different SEB components, allowing readers to discern relative contributions effectively. The inclusion of panel (b), depicting composite absolute AR-

related SEB term anomalies adjacent to original panel (c), serves to remind readers to consider absolute anomaly values alongside relative contributions. Presenting both the anomaly (panel (b)) and relative contribution (original panel (c)) aims to provide readers a comprehensive perspective, highlighting terms that are large in both absolute and relative senses (e.g., downward longwave radiation over sea ice-covered central Arctic Ocean), as well as those that are small in absolute anomalies but substantial relative to the overall surface energy budget (e.g., SEB terms over continents).

We appreciate the Reviewer's suggestion to show each AR SEB term normalized by just the mean of that term and think that this is a very useful suggestion. As such we have included this with a slight modification as an additional panel (now panel (c)) in Figures 2, 3 and 5-7 and moved the AR SEB contribution normalized by the total SEB (original panel (c)) to panel (d). We have included the equations used to calculate these results of new panel (c) and (d) in Section 2 (Data and Methods) for transparency and clarity in the manuscript, as follows:

"Mathematically, the results shown in panel (c) result from the following calculation at each individual grid point within the study domain for each season:

- I. Calculate the total extra energy contributed by each SEB component when ARs are present as,  $(F_{AR} F_{All}) * t_{AR}$ , where  $F_{AR}$  represents the mean of any term in the SEB equation when an AR is present,  $F_{All}$  denotes the seasonal mean of any term in the SEB equation (panel (a)), and  $t_{AR}$  indicates the total number of 3-hourly time steps during which ARs are present.
- II. Calculate the total energy for each component as,  $F_{All} * t_{All}$ , where  $t_{All}$  signifies the total number of 3-hourly time steps within each season.
- III. Determine the ratio of these two terms, which provides an estimate of the magnitude of AR anomaly for each SEB term relative to the average value for each component. This is presented in Eq. (2), noting that the ratio of  $t_{AR}$  to  $t_{All}$  is simply the AR frequency shown in Fig. 1.  $\frac{(F_{AR}-F_{All})*t_{AR}}{(p_{AR}-F_{All})*t_{AR}} = \frac{panel(b)*Fig.1}{(p_{AR}-F_{All})*t_{AR}} = \frac{panel(b)}{(p_{AR}-F_{All})} = \frac{panel(b)$

$$\frac{F_{AR} - F_{All} * t_{AR}}{F_{All} * t_{All}} = \frac{panel (b) * Fig.1}{panel (a)}$$
(2)

Furthermore, the results depicted in panel (d) stem from the following calculation conducted at each individual grid point within the study domain for each season.:

- *I.* Calculate the total extra energy contributed by each term in the SEB equation when ARs are present as:  $(F_{AR} F_{All}) * t_{AR}$
- *II.* Compute the absolute value of total SEB energy as:  $|netSEB_{All}| * t_{All}$ , where  $|netSEB_{All}|$  represents the absolute value of seasonal mean net SEB at a given grid point.
- *III.* The ratio of these two terms indicates the relative contribution of the AR anomaly for each SEB term to the total seasonal SEB, as shown in Eq (3).

 $\frac{(F_{AR}-F_{All})*t_{AR}}{|net SEB_{All}|*t_{All}} = \frac{panel (b)*Fig.1}{|Fig.7(a)|}$ (3)"

The alternative solution presented in Equation 6 would yield results equivalent to  $\frac{(LWD_{noAR} - LWD_{AR})*t_{AR}}{SEB_{All}*t_{All}} = \frac{(LWD_{noAR} - LWD_{AR})*\frac{t_{AR}}{t_{All}}}{SEB_{All}}.$ This alternative approach also involves the net SEB as the denominator in the calculation, resulting in results greater/less than 100%/ -100%. After careful consideration, we decide to add the metric of each AR SEB term normalized by just the mean of that term (new panel (c)) and maintain our original metric of AR SEB contribution normalized by the total SEB (now panel (d)). The corresponding specific equations are provided in the manuscript, as stated above.

2. I am unable to reproduce the "contribution to SEB" values shown in Table 1 using the description of how it was calculated in Section 2.3. Since the AR frequencies, anomalies and net SEB values are provided for each region, the contribution should be able to be calculated without any further information (based on Section 2.3). Please see the attached document for an example of this calculation not resulting in the same value seen in Table 1.

Reply: We apologize for any lack of clarity in our methodology section. For each SEB term, we summarize key metrics in Table 1 and Table S1, such as AR occurrence frequency (Fig.1), climatology (panel a), composite anomalies (panel b), AR contribution to individual SEB component (panel c), and total AR contribution to absolute net SEB (now panel d). These results listed in the tables are derived from area-averaged calculations, which involves summing the results of grid points falling within each region and weighing them using the cosine values of their corresponding latitudes. Mathematically, the weighted average of a metric f over a grid with latitude  $\theta$  can be represented as:

$$< f >= \frac{\sum_{i=1}^{n} w_i f_i}{\sum_{i=1}^{n} w_i}$$

where  $\langle f \rangle$  represents the weighted average of the metric f,  $f_i$  is the value of f at grid point i,  $w_i$  is the weight associated with the grid point i, which is the cosine of the latitude at that point,  $(w_i = \cos(\theta_i))$ , and n is the total number of grid points within each region.

The results of the relative contribution of AR LWD to absolute net SEB listed in Table 1 are calculated as bellow.

1. For each grid point *i* within a specific region during each season, we calculate the metric of  $f_i$  using Eq. (2):

$$f_i = \frac{(F_{AR,i} - F_{All,i}) * \frac{t_{AR,i}}{t_{All,i}}}{|net SEB_{All,i}|} = \frac{panel (b)_i * Fig. 1_i}{|Fig. 7(a)|_i}$$

2. We then multiply the value of  $f_i$  by the weight  $w_i$  ( $w_i = \cos(\theta_i)$ ), resulting in  $f_i * \cos(\theta_i)$

- 3. We sum up these products of  $f_i * \cos(\theta_i)$  for all the grid points (n) within each region as  $\sum_{i=1}^{n} f_i * \cos(\theta_i)$
- 4. This sum is divided by the total sum of weights:  $\langle f \rangle = \frac{\sum_{i=1}^{n} f_i * \cos(\theta_i)}{\sum_{i=1}^{n} \cos(\theta_i)}$

However, it is important to note that the weighted aera-averaged result of the relative contribution to net SEB using all the grid points within each season is not equivalent to the result calculated directly using the weighted average results of composite anomalies (< Fig. 2b >), AR frequency (< Fig. 1 >) and net SEB (< Fig. 7a >) listed in the table. For example, the weighted average of composite LWD anomalies (Fig.2b) can be written as:

$$\langle Fig. 2b \rangle = \frac{\sum_{i=1}^{n} Fig. 2b_i * \cos(\theta_i)}{\sum_{i=1}^{n} \cos(\theta_i)}$$

Similarly, the weighted average of AR occurrence frequency (Fig. 1) and net SEB (Fig. 7a) are calculated as:

$$\langle Fig.1 \rangle = \frac{\sum_{i=1}^{n} Fig. 1_i * \cos(\theta_i)}{\sum_{i=1}^{n} \cos(\theta_i)}$$

$$\langle Fig.7a \rangle = \frac{\sum_{i=1}^{n} Fig.7a_i * \cos(\theta_i)}{\sum_{i=1}^{n} \cos(\theta_i)}$$

The weighted average of AR LWD contribution to absolute of net SEB can be written as:

$$\langle Fig. 2d \rangle = \frac{\sum_{i=1}^{n} \frac{Fig. 2b_i * Fig1_i}{|Fig. 7a_i|} * \cos(\theta_i)}{\sum_{i=1}^{n} \cos(\theta_i)}$$

However, it is important to highlight that  $\sum_{i=1}^{n} \frac{Fig.2b_i * Fig1_i}{|Fig.7a_i|} * cos\theta_i$  is not simply equivalent to

 $\frac{(\sum_{i=1}^{n} Fig.2b_{i}*cos\theta_{i})*(\sum_{i=1}^{n} Fig.1_{i}*cos\theta_{i})}{\sum_{i=1}^{n} |Fig.7a_{i}|*cos\theta_{i}}$ Thus,  $< Fig.2d > \neq \frac{<Fig.2b>*<Fig.1>}{<Fig.7a>}$

To illustrate, I provide a specific example of a few values to calculate the AR-related LWD contribution to the total net SEB, as mentioned by the Reviewer:

| longitude              | point | weighted |
|------------------------|-------|-------|-------|-------|-------|-------|-------|-------|----------|
| (170 °E)               | i=1   | i=2   | i=3   | i=4   | i=5   | i=6   | i=7   | i=8   | results  |
| Latitude (°N)          | 70    | 75    | 78    | 74    | 76    | 80    | 82    | 86    |          |
| LWD AR      | 29.8  | 30.1  | 32.3  | 28.7  | 31.7  | 31.5  | 31.7  | 32.4  | 30.7     |
| $-LWD_{All}(W m^{-2})$ |       |       |       |       |       |       |       |       |          |

| $\frac{t_{AR}}{t_{All}} (\%)$                                          | 12.0 | 11.4  | 10.7  | 11.8  | 11.2  | 10.5  | 9.7   | 8.4   | 11.1  |
|------------------------------------------------------------------------|------|-------|-------|-------|-------|-------|-------|-------|-------|
| net SEB ( $W m^{-2}$ )                                                 | -8.3 | -14.8 | -15.8 | -14.4 | -15.4 | -16.7 | -17.8 | -19.1 | -14.3 |
| $\frac{(F_{AR}-F_{All})*\frac{t_{AR}}{t_{All}}}{ net SEB_{All} } (\%)$ | 43.1 | 23.2  | 21.9  | 23.5  | 23.1  | 19.8  | 17.3  | 14.2  | 25.9  |

Based on this example with just 8 grid points, the area-averaged result of the relative AR contribution to the net SEB is 25.9%; while directly using the weighted results will lead to the relative contribution of 30.7\*11.1/14.3=23.8%. In fact, the total number of grid points for the four sub-regions range from 51844 to 85175, leading to much greater differences in the results using the two approaches mentioned before. The calculation of area-averaged results weighted by the cosine of latitudes accounts for the convergence of meridians towards the poles, ensuring a more accurate representation of the area-averaged results. Additionally, we have included the methods to calculate the results listed in the table in Section 2.3 for clarity, as follows:

"We summarize key features from Figures 2-3,5-7 into Table 1 and Table S1 to analyze each SEB component and the net SEB across four sub-regions: the central Arctic (including the Barents and Kara Seas), sub-polar oceans, continents, and Greenland (Fig. S1), for every season. These tables present regional averages for several metrics, including climatology (panels a), composite anomalies (panels b), AR contribution to individual SEB component (panels c), AR contribution to absolute net SEB (panels d), along with AR frequency (as shown in Fig.1). To derive these results, we perform area-averaged calculations by summing the values from grid points within each region and weighting them based on the cosine values of their corresponding latitudes. Additionally, we calculate the difference between the area-averaged AR contribution to the net SEB and the area-averaged AR frequency, representing additional AR contribution, which is presented in the last row of the tables."

3. Consider performing statistical testing to determine if the absolute anomalies during ARs are statistically different from the mean conditions (which could be shown in the b rows of Figures 2-7). Since ARs exist in a location likely for more than one timestep, there is some temporal autocorrelation which may be accounted for by randomly selecting a smaller sample of AR timesteps to compare to a randomly selected sample of non-AR timesteps. Determining the statistical significance of these anomalies may help to identify SEB components that are more important with more confidence.

Reply: While evaluating the statistical significance of these anomalies is indeed valuable for identifying AR-related important SEB components with greater confidence, we have concerns that labelling all the grid points within the confidence intervals may potentially overwhelm readers and make it hard to read. Therefore, we have decided to conduct statistical confidence intervals for the regional area-averaged results listed in Table 1 and Table S1 using the bootstrap method. We will indicate the results within the 95% (or 90%) confidence intervals by presenting

them in bold text and adding them to the table caption. This approach aims to maintain clarity while still conveying the statistical confidence of our findings effectively.

**Minor Comments:**

45-46: ARs typically being associated with extratropical cyclones is mentioned here, but isn't discuss it again. I think more discussion regarding the linkage between cyclones and ARs would be valuable here for context of when/how ARs occur in the Arctic.

Reply: We have addressed your comments by expanding the discussion on the relationship between ARs and cyclones, particularly focusing on their association in the Arctic region, as follows:

"Atmospheric rivers (ARs) are long and narrow filaments of enhanced moisture transport typically associated with a low-level jet and extratropical cyclone (Ralph et al., 2018). In mid-latitudes, ARs are commonly identified in the warm conveyor belts of synoptic-scale cyclones, particularly low-level jets (Ralph et al., 2004, 2006). Some literature even considers ARs as part of cyclones (Bao et al., 2006; Neiman et al., 2008; Dacre et al., 2015). ARs and cyclones exhibit strong statistical and dynamic relationships (Zhang et al., 2019; Guo et al., 2020; Eiras-Barca et al., 2018). In the Arctic, poleward moisture transport is also closely linked to cyclone activity, including intensity, frequency, and duration (Villamil-Otero et al., 2018). Arctic cyclones account for over 70% of the average annual moisture transport, with their track orientation and upperlevel steering flow significantly influencing poleward moisture flux (Fearon et al., 2021)."

100: It is mentioned that times are only used during neutral or weak El Niño-Southern Oscillation. I assume it's because of IVT anomalies associated with strong ENSO events, but it is worth briefly stating in the text for clarity.

Reply: We only preserved the dates during neutral or weak ENSO events to establish a standard climate threshold for testing ARs. For instance, in the event of updating the AR index to the MOSAiC year, we do not need to collect the data and recalculate the thresholds. To clarify this approach, we have included the following note in the manuscript:

"The selection of the neutral or weak ENSO events aim to establish a standard climate threshold for testing ARs."

123-125: Is it necessary to give multiple names for these first 3 ERA5 variables? Reply: The term "surface thermal radiation downward", "surface net thermal radiation", and "surface net solar radiation" are the names provided in the original ERA5 source data. However, in the literature, the commonly used terms are "surface downward longwave radiation", "surface net longwave radiation", and "surface net shortwave radiation". We choose to use these commonly used names while also mentioning their origins in ERA5 for the benefit of readers who wish to replicate the results.

147-149: This sentence uses both "three-hourly" and "3-hourly" referring to the data – I suggest picking one to remain consistent.

Reply: We have selected "3-hourly" and replaced all instances of "three-hourly" to "3-hourly" throughout the manuscript.

287-289: What is meant by "ARs make their most significant relative contribution to the average net SEB in spring, accounting for at least 45% of the net SEB, surpassing the corresponding AR frequency by more than 34%"? I don't think subtracting the frequency from the contribution has a physical meaning since they are percentages of different things.

Reply: The difference between the area-averaged AR contribution to the net SEB and the area-averaged AR frequency, indicates additional AR contribution. The difference signifies the extent to which ARs contribute to the net SEB beyond their occurrence frequency.

358: I suggest starting a new paragraph at "The results over the central Arctic" as this is a long paragraph, and a new topic is being introduced here. Reply: We have started a new paragraph for this topic.

Section 3 is titled "Analysis and Results" and Section 4 "Discussion", but Section 3 includes a lot of discussion (i.e., discussing potential impacts of the anomalies, comparing to previous work) and Section 4 still discusses some results (particularly temperature). A potential solution for this would be to rename Section 3 to focus on SEB and Section 4 to focus on impacts, and perhaps create another section for limitations/uncertainties (for 4.3 and 4.4)

Reply: We have taken this suggestion into account and restructured the manuscript accordingly. Therefore, the sections have been retitled as per the suggestion.

References

Bao, J. W., Michelson, S. A., Neiman, P. J., Ralph, F. M., and Wilczak, J. M.: Interpretation of enhanced integrated water vapor bands associated with extratropical cyclones: Their formation and connection to tropical moisture, Mon Weather Rev, 134, https://doi.org/10.1175/MWR3123.1, 2006.

Dacre, H. F., Clark, P. A., Martinez-Alvarado, O., Stringer, M. A., and Lavers, D. A.: How do atmospheric rivers form?, Bull Am Meteorol Soc, 96, https://doi.org/10.1175/BAMS-D-14-00031.1, 2015.

Eiras-Barca, J., Ramos, A. M., Pinto, J. G., Trigo, R. M., Liberato, M. L. R., and Miguez-Macho, G.: The concurrence of atmospheric rivers and explosive cyclogenesis in the North Atlantic and North Pacific basins, Earth System Dynamics, 9, https://doi.org/10.5194/esd-9-91-2018, 2018.

Fearon, M. G., Doyle, J. D., Ryglicki, D. R., Finocchio, P. M., and Sprenger, M.: The Role of Cyclones in Moisture Transport into the Arctic, https://doi.org/10.1029/2020GL090353, 2021.

Guo, Y., Shinoda, T., Guan, B., Waliser, D. E., and Chang, E. K. M.: Statistical relationship between atmospheric rivers and extratropical cyclones and anticyclones, J Clim, 33, https://doi.org/10.1175/JCLI-D-19-0126.1, 2020.

Neiman, P. J., Ralph, F. M., Wick, G. A., Lundquist, J. D., and Dettinger, M. D.: Meteorological characteristics and overland precipitation impacts of atmospheric rivers affecting the West coast of North America based on eight years of SSM/I satellite observations, J Hydrometeorol, 9, https://doi.org/10.1175/2007JHM855.1, 2008.

Ralph, F. M., Neiman, P. J., and Wick, G. A.: Satellite and CALJET aircraft observations of atmospheric rivers over the Eastern North Pacific Ocean during the winter of 1997/98, Mon Weather Rev, 132, https://doi.org/10.1175/1520-0493(2004)132<1721:SACAOO>2.0.CO;2, 2004.

Ralph, F. M., Neiman, P. J., Wick, G. A., Gutman, S. I., Dettinger, M. D., Cayan, D. R., and White, A. B.: Flooding on California's Russian River: Role of atmospheric rivers, Geophys Res Lett, 33, https://doi.org/10.1029/2006GL026689, 2006.

Ralph, F. M., Dettinger, M. C. L. D., Cairns, M. M., Galarneau, T. J., and Eylander, J.: Defining "Atmospheric river" : How the glossary of meteorology helped resolve a debate, Bull Am Meteorol Soc, 99, https://doi.org/10.1175/BAMS-D-17-0157.1, 2018.

Villamil-Otero, G. A., Zhang, J., He, J., and Zhang, X.: Role of extratropical cyclones in the recently observed increase in poleward moisture transport into the Arctic Ocean, Adv Atmos Sci, 35, https://doi.org/10.1007/s00376-017-7116-0, 2018.

Zhang, Z., Ralph, F. M., and Zheng, M.: The Relationship Between Extratropical Cyclone Strength and Atmospheric River Intensity and Position, Geophys Res Lett, 46, https://doi.org/10.1029/2018GL079071, 2019.

---

## Referee Report (RR1)

**Review for The Cryosphere: Quantifying the Impacts of Atmospheric Rivers on the Surface Energy Budget of the Arctic Based on Reanalysis Review #2**

1. The addition of the new panel C in Figures 2-3 and 5-7 is appreciated and helps illustrate how important ARs are to each term of the surface energy budget. However, the original metric still does not seem mathematically sound enough for the conclusions that are being drawn from it.

In the SEB contribution metric, the relative importance is inflated in some regions since the net SEB is near zero. Therefore, even a very small anomaly (e.g.,  $< 8 \text{ W m}^{-2}$ ) is deemed to have a massive contribution even if it isn't enough to make any kind of material difference to the SEB/temperature/etc. This is especially true over land regions where seasonal net SEB can be very small (e.g., SEB =  $0.4 \text{ W m}^{-2}$  for Greenland in MAM). This is an especially problematic issue because some of the terms contributing to the SEB are of higher magnitude than the net SEB, with correlated but offsetting terms (e.g., LWD v. LWU or LWnet v. SWnet) can make the SEB near zero seasonally. Because of this, you can have anomalies that seem large when compared to net SEB, but actually are only a small contributor to seasonal totals (which is illustrated well in Figure 2 panels c vs d).

If you consider the spatial pattern of net seasonal SEB, it is evident that the spatial variation in the relative contribution results (e.g., Figure 2d) are determined primarily by variation in the denominator (net SEB) rather than the numerator (e.g., LWD, LWnet, etc.). For example, in Figure 2d, there are much higher relative values over land than over the ocean, especially in Fall and Winter, least so in Summer. This tracks exactly with the net SEB being practically 0 W/m2 over land in Fall and Winter and appreciably positive (downward) in Summer. Every variable (including some that counteract each other) gives higher relative results over the continents. This issue is well demonstrated by Spring in Figures 2 (LWD) and 3 (LWnet). Comparing central Siberia to the Laptev Sea, LWD anomalies are clearly stronger over land than over the Laptev Sea (Figure 2b). In relative terms, the continental anomalies are also higher than the Laptev Sea (Figure 2d). Both central Siberia and the Laptev Sea have smaller absolute anomalies in LWnet than LWD – and the anomalies about the same continent versus land (Figure 3b). However, the relative impact again shows the continental anomalies being higher. It doesn't matter if the land has higher magnitude absolute anomalies than the ocean or not – both Figure 2 and Figure 3 yield higher relative anomalies for the land. In other words, the relative figures are telling us a lot more about the denominator than the numerator, meaning Figure 2d and Figure 3d are primarily telling us about SWnet rather than LWD or LWnet.

Caution should be used when drawing conclusions from a metric that can be easily skewed. The relative metric used is not mathematically incorrect, but the issue lies in how strongly the conclusions are drawn from said metric. For example, lines 914-916: "ARs generate relatively smaller absolute anomalies in net SEB over continental areas (Fig. 7b), mainly attributable to the LWN. However, their impact on the mean SEB is substantial, especially in cold seasons (24-90%), far exceeding the corresponding frequency, which is primarily due to the smaller mean SEB (Fig. 7a)." Here, it is stated that the "large contribution" is really just because of the small net SEB. However, the relative contribution metric is still used consistently to mathematically compare to the frequency and draw the main conclusions (e.g., in the abstract), even though it is clear that it is easily skewed.

The goal of understanding AR's contribution to seasonal net SEB can even be well approximated by directly comparing the mean absolute anomalies during ARs to the mean seasonal conditions for each term and the net SEB. Many of the conclusions can still be made supported by the values shown in panels (b) and (c), rather than strongly relying on panel (d) for the main conclusions. Therefore, although calculating a relative anomaly is mathematically fine (and they can stay in the results), relying on such anomalies in this case is logically flawed, and the authors should change the focus of the conclusions and abstract to reflect this.

2. I agree with other reviewers that the net SEB (Figure 7) should be shown much earlier (even before Figure 2 since the results in Figure 2 are relative to the net SEB). It would be helpful to illustrate the regions of low net SEB earlier so it's easy to see why those >> 100% contributions exist.

**Technical comments:**

- Fig 2: the colourbars for panels (c) and (d) are very close to being the same but are slightly different – suggest using the same colourbar if they're almost identical to make it easier to interpret
- Line 338: "the modest large contributions" unclear how the contribution is both modest and large (perhaps "the modest increase in contributions" is what is meant here)
- Line 474-475: "except in winter when reduced climatological LWN cooling leads to a slight increase contribution" → "a slight increase in contribution"
- Supplemental figures: for the statistical significance figures, "The grey dots are plotted over regions of anomalies outside of the 95% confidence intervals based on two-tailed t-test" is unclear Are the dots shown where the anomalies are not statistically significant at the 95% level?

---

## Author Response (AR2)

egusphere-2024-320

"Quantifying the Impacts of Atmospheric Rivers on the Surface Energy Budget of the Arctic Based on Reanalysis"

**Response to the Reviewers**

**Second Revision Response to the Reviewers**

**By Chen Zhang, John J. Cassano, Mark Seefeldt, Hailong Wang, Weiming Ma, and Wenwen Tung**

We appreciate that the inclusion of the new metric and corresponding results shown in panel (c) were well received by the reviewers in the first round of revisions. In the following, we respond point-by-point replies to each reviewer's comments in blue, using an italic font to indicate text that has been copied verbatim from the Reviewer's reports. We also supplement this revised submission with a copy of the manuscript in which changes from the previous version are highlighted in red.
* * *
**Reply to Reviewer #2,** Jonathan Wille:**

We thank the reviewer for the detailed and insightful comments in the first round that improved our manuscript. We are glad that the reviewer found our answers satisfactory. We have also made all suggested changes this time.

Reviewer #2: Jonathan Wille

I am happy with the response to my comments along with the response to the other reviewers who made important points regarding the study's methodology. I have one technical comment related to the added text.

Line 605-606: I find this sentence confusing. How can positive AR anomalies in turbulent heat "contribute only moderately negatively to TH climatology..."? Shouldn't the AR anomalies contribute positively or make the TH climatology slightly less negative?

Reply: Thank you for pointing it out. We have changed to "Over subpolar oceans, substantial positive AR TH anomalies act to partially offset the strong negative TH climatology, averaging a -7% to -11% relative to its climatology".

\_\_\_\_\_

**Reply to Reviewer #3:**

We truly appreciate the Reviewer's detailed and thoughtful critique for further clarification.

**Reviewer #3:**

1. The addition of the new panel C in Figures 2-3 and 5-7 is appreciated and helps illustrate how important ARs are to each term of the surface energy budget. However, the original metric still does not seem mathematically sound enough for the conclusions that are being drawn from it.

In the SEB contribution metric, the relative importance is inflated in some regions since the net SEB is near zero. Therefore, even a very small anomaly (e.g., < 8 W m-2) is deemed to have a massive contribution even if it isn't enough to make any kind of material difference to the SEB/temperature/etc. This is especially true over land regions where seasonal net SEB can be very small (e.g., SEB = 0.4 W m-2 for Greenland in MAM). This is an especially problematic issue because some of the terms contributing to the SEB are of higher magnitude than the net SEB, with correlated but offsetting terms (e.g., LWD v. LWU or LWnet v. SWnet) can make the SEB near zero seasonally. Because of this, you can have anomalies that seem large when compared to net SEB, but actually are only a small contributor to seasonal totals (which is illustrated well in Figure 2 panels c vs d).

If you consider the spatial pattern of net seasonal SEB, it is evident that the spatial variation in the relative contribution results (e.g., Figure 2d) are determined primarily by variation in the denominator (net SEB) rather than the numerator (e.g., LWD, LWnet, etc.). For example, in Figure 2d, there are much higher relative values over land than over the ocean, especially in Fall and Winter, least so in Summer. This tracks exactly with the net SEB being practically 0 W/m2 over land in Fall and Winter and appreciably positive (downward) in Summer. Every variable (including some that counteract each other) gives higher relative results over the continents. This issue is well demonstrated by Spring in Figures 2 (LWD) and 3 (LWnet). Comparing central Siberia to the Laptev Sea, LWD anomalies are clearly stronger over land than over the Laptev Sea (Figure 2b). In relative terms, the continental anomalies are also higher than the Laptev Sea (Figure 2d). Both central Siberia and the Laptev Sea have smaller absolute anomalies in LWnet than LWD – and the anomalies about the same continent versus land (Figure 3b). However, the relative impact again shows the continental anomalies being higher. It doesn't matter if the land has higher magnitude absolute anomalies than the ocean or not – both Figure 2 and Figure 3 yield higher relative anomalies for the land. In other words, the relative figures are telling us a lot more about the denominator than the numerator, meaning Figure 2d and Figure 3d are primarily telling us about SWnet rather than LWD or LWnet.

Caution should be used when drawing conclusions from a metric that can be easily skewed. The relative metric used is not mathematically incorrect, but the issue lies in how strongly the conclusions are drawn from said metric. For example, lines 914-916: "ARs generate relatively smaller absolute anomalies in net SEB over continental areas (Fig. 7b), mainly attributable to the LWN. However, their impact on the mean SEB is substantial, especially in cold seasons (24-90%), far exceeding the corresponding frequency, which is primarily due to the smaller mean SEB (Fig. 7a)." Here, it is stated that the "large contribution" is really just because of the small net SEB. However, the relative contribution metric is still used consistently to mathematically compare to the frequency and draw the main conclusions (e.g., in the abstract), even though it is clear that it is easily skewed.

The goal of understanding AR's contribution to seasonal net SEB can even be well approximated by directly comparing the mean absolute anomalies during ARs to the mean seasonal conditions for each term and the net SEB. Many of the conclusions can

still be made supported by the values shown in panels (b) and (c), rather than strongly relying on panel (d) for the main conclusions. Therefore, although calculating a relative anomaly is mathematically fine (and they can stay in the results), relying on such anomalies in this case is logically flawed, and the authors should change the focus of the conclusions and abstract to reflect this.

Response: We sincerely appreciate the Reviewers' detailed and insightful comment regarding the interpretation of the relative contribution metric to the net SEB (panels (d)). We agree that the results of the relative metric to net SEB (panels (d)) is primarily influenced more by the magnitude of the net SEB itself-particularly over continental regions and in cold seasons where the climatological net SEB is near 0- rather than solely by the magnitudes of term anomalies (panels (b)). As the Reviewer rightly points out, this can lead to disproportionately large relative values, which may give a misleading impression of physical significance of AR impacts.

In response, we have taken the following steps:

**1. Clarified and revised interpretation in main text.**

We have revised our interpretations throughout key sections (e.g., 4.1-4.3) to reflect a more cautious interpretation. Specifically, we now explicitly state that the high relative values in panel (d) are largely driven by small background of net SEB values, especially over land in cold reasons, and do not necessarily indicate large absolute AR-induced SEB term anomalies.

**2. Revised Abstract and Conclusions sections.**

We have updated the Abstract and Conclusions to shift the emphasis away from the relative net SEB contribution metric (panel d). Instead, we highlight the absolute anomalies and their relative contribution to the corresponding climatology (panels b and c), which provide a more physically direct interpretation of AR impacts. We also include a statement of caution regarding the use of relative metric, clarifying that it should be interpreted in context and not isolation to draw conclusions. We also emphasize that both absolute anomalies and relative contributions, both to climatological means and to net SEB, are complementary and together provide a more comprehensive understanding of AR-induced surface energy budget changes.

**3. Panels (d) with adjusted framing**

We continue to keep the panels (d) in the main figures and corresponding discussion since the goal of the manuscript is to estimate the relative contribution of different SEB components to the net SEB across the Arctic. However, the results of panel (d) are no longer used as the main basis for our key conclusions, but as the supplementary metric.

2. I agree with other reviewers that the net SEB (Figure 7) should be shown much earlier (even before Figure 2 since the results in Figure 2 are relative to the net SEB). It would be helpful to illustrate the regions of low net SEB earlier so it's easy to see why those >> 100% contributions exist.

Response: We appreciate the Reviewer's point and now agree that presenting the net SEB earlier can help contextualize the contribution of individual SEB components to the net SEB (panel d). In response, we have moved the net SEB figure earlier in the manuscript (now shown in Figure 3).

in the manuscript) to help readers visualize the spatial distribution for the rest of the analysis. While we did not place it before Figure 2, we adjusted its position in accordance with the logical flow and figure order mentioned in the manuscript.

While the figure itself has been moved forward, we retain the in-depth discussion of the net SEB (now Figure 3) in its original location to proceed through other individual terms in the SEB first and ending with the net SEB, which sums the previously discussed results. All corresponding figure references and related text have been updated to reflect this reordering. We hope this reorganization improves the overall clarity and flow of our discussion.

**Technical comments:**

• Fig 2: the colourbars for panels (c) and (d) are very close to being the same but are slightly different – suggest using the same colorbar if they're almost identical to make it easier to interpret

Response: Thank you very much for your thoughtful suggestion. We agree that visual consistency between color bars can enhance interpretability. However, while the color bars for panels (c) and (d) are similar in structure, they are intentionally not identical due to the difference in the magnitude of the values displayed. Panel (c) shows much smaller magnitudes than panel (d) and using the exact same scale would obscure important spatial variations in panel (c). To improve clarity and comparability, we designed the color bars such that the scale from 0% to 4% is subdivided more finely in panel (c), allowing key features to be more visible. Beyond 4%, both panels share the same scale. We believe this approach balances the need for consistency with the need to clearly visualize the distinct features in each panel.

• Line 338: "the modest large contributions" – unclear how the contribution is both modest and large (perhaps "the modest increase in contributions" is what is meant here)

Response: Fixed it.

- Line 474-475: "except in winter when reduced climatological LWN cooling leads to a slight increase contribution"→ "a slight increase in contribution" Response: Fixed it.
- Supplemental figures: for the statistical significance figures, "The grey dots are plotted over regions of anomalies outside of the 95% confidence intervals based on two-tailed t-test" is unclear Are the dots shown where the anomalies are not statistically significant at the 95% level

Response: We have revised the description to "The grey dots indicate regions where the anomalies are not statistically significant at the 95% confidence level, based on a two-tailed t-test." We hope this updated wording clarifies the intended meaning.

**Reply to Reviewer #4:**

We express our gratitude to Reviewer#4 for their insightful and constructive feedback aimed at enhancing our manuscript.

Reviewer #4:

This study quantitatively assessed the contributions of AR on the Arctic surface energy budget for four decades using ERA2 and MERRA 2 reanalysis data. The analysis noted that the ARs resulted in positive LWD anomalies in all seasons, particularly strong in transition seasons, which positively contributed to the net SEB. The AR impacts were also linked to the Arctic sea ice retreat, particularly in sea ice boundary regions, and Greenland glacier melting, which is likely a main result of this study compared to the AR related previous studies.

Response: We appreciate the Reviewer's thoughtful summary of our study. However, we would like to clarify that the current study is based solely on the ERA5 reanalysis date. We do not use

ERA-Interim (ERA2) or MERRA-2 in this analysis. We hope this clarification helps avoid

confusion regarding the data sources.

This study highlighted the impact of the AR induced energy budget anomalies on sea ice and Greenland glacier melting. The relationship between ARs and Arctic cryospheric changes, particularly in the Arctic Ocean is certainly an interesting topic in climate change researches, and the resultant findings will highly contribute to understand the arctic amplification as well as the future projection. Here it is questioned on why this study was mainly focused on the Arctic Ocean. For example, the AR certainly resulted in positive LWD anomalies in all seasons, while the contribution of the anomalies on the SEB components was overwhelmingly larger in land than ocean. Of course, the manuscript shortly included the point. However, the AR impact related description and discussion was almost directed to the sea ice and glacier. In the cold season when sea ice is formed, grown, and melted in the ocean, snow cover is simultaneously appeared and disappeared on land surface, as well as permafrost is also freezing and warming. The AR induced changes in SEB likely affect the land snow and permafrost, because which are largely implicated to the Arctic climates and arctic amplification, having impacts equivalent to the sea ice change. Therefore, you have also equally to deal with the impacts on the land in the manuscript.

Response: We sincerely thank the Reviewer for this insightful comment highlighting the importance of considering AR-induced energy budget changes over land surfaces. We agree that AR-driven SEB anomalies have the potential to significantly impact snow cover and permafrost, especially during cold seasons, and that these land-based cryosphere changes play an equally important role in Arctic amplification as sea ice and glacier responses. In response to this helpful suggestion, we have expanded the corresponding section to more thoroughly address the potential implications of AR-induced SEB terms anomalies for snow cover impacts and permafrost stability.

For example, regarding the discussion of LWD, we include the statement in Lines 319-321: "These AR-related LWD anomalies are particularly important over continental regions where they may contribute to the warming and thawing of seasonally snow-covered ground and permafrost, potentially altering surface energy budgets and hydrological process (Guan et al., 2016; Goldenson et al., 2018)."

The Lines 445-447 are included for the LWN: "While smaller in magnitude, the AR-induced LWN anomalies also appear over Arctic continental regions. In cold seasons, the anomalous LWN from ARs can contribute to earlier snowmelt, reduced snow accumulation, and altered permafrost thermal regimes"

In the Conclusions section, we also expand on that to acknowledge the broader relevance of our findings to Arctic land processes and climate feedback. Such as in Line 978-980: "Nonetheless, even moderate AR-driven SEB perturbations might have significant impacts on Arctic continental environments, potentially accelerating permafrost thaw, altering snowpack evolution, or affecting surface hydrology, ultimately shaping land-surface processes."

The ARs induced higher LWD also resulted in higher SST, which increased outward longwave radiation, consequently lower LWN, which was in turn offset by the negative SWN caused by the ARs. Based on a simple budget calculation, the energy source that likely forced the sea ice and glacier melting was probably TH anomalies from the atmosphere. The resultant net SEB anomalies caused by the ARs were less than approximately 20 W/m2 in the sea ice covered ocean. It is questioned on how much the 20 W/m2 can derive sea ice melting. For instance, in spring that sea ice starts to melt, snow over the sea ice is melted earlier than the sea ice melting. These process suggests that the AR induced net SEB may be almost used for the snow melting, although it depends on snow amounts. Therefore, the assertation that the ARs induced net SEB likely triggered the sea ice and glacier melting is an overstated expression. If the net SEB affected the sea ice and glacier, the influence was likely limited to the sea ice margins, particularly North Atlantic, Barents, and Chukchi sea where SST was relatively warm. This was included in the manuscript, but the description could cause a misunderstanding that the ARs likely affected the sea ice melting of the Arctic scale.

Response: We sincerely appreciate the Reviewer's thoughtful critique and detailed discussion regarding the interpretation of AR-induced net SEB anomalies and their potential role in sea ice and glacier melt. We agree that caution is warranted when attributing sea ice and glacier melt directly to AR-induced net SEB anomalies, especially given the complex processes involved in the seasonal melt transition.

In response, we have revised the relevant statements in the revised manuscript to avoid overstatement. Specifically:

Lines 325-326 (regarding AR-related LWD anomalies): "However, the degree to which these anomalies contribute to sea ice melt or delay ice growth likely depend on local cryosphere conditions, such as snow and ice conditions."

Lines 696-699 (regrading AR-related net SEB anomalies): "However, the extent to which these anomalies directly trigger sea ice melt likely depends on local snow and ice conditions, and the influence is expected to be most relevant in regions with relatively warmer sea surfaces such as the Barents, Chukchi, and Arctic suboceanic sectors. In areas with thicker snow cover or colder background conditions, these anomalies may have more limited or indirect effects."

Lines 938-941 in the Conclusions Section: "These short-term increases in the SEB potentially hinder sea ice refreezing in fall and winter (Zhang et al, 2023b) and may trigger sea ice melt in spring (Huang, Dong, Bailey, et al., 2019; Huang, Dong, Xi, et al., 2019). Nonetheless, the impact of these anomalies on initiating sea ice melt is contingent upon regional snow and ice conditions, with the most pronounced regions characterized by warm sea surface temperatures"

Line 947-949 in the Conclusions Section: "In spring, the combination of large AR-induced LWD anomalies and a smaller net SEB climatology result in a substantial relative contribution to the climatological net SEB. This suggests that AR-driven SEB anomalies play an important role in modifying the climatological SEB, potentially supporting early-state melt in spring, particularly of snow over sea ice, and influencing the minimum sea ice extent in fall (Huang et al., 2019)."

This study was based on seasonal means during the period 1979–2019. If the AR induced anomalous energy budgets affected the sea ice and glacier melting, the impacts could be more significant at the recent decade when the climates were relatively warm. That is, the anomalous impacts include larger uncertainty in the interannual variability. Furthermore, the seasonal means have also the similar uncertainty. For example, in spring when the ARs induced net SEB anomalies were relatively large, if the AR events were biased to the early spring of colder climates, the impacts on the sea ice and glacier could not represent the seasonal variability. Of course, there are the contrast cases.

Response: We thank the Reviewer for this important comment regarding the variability of AR-induced anomalous energy budgets and their impacts on sea ice and glacier melt. We agree that interannual variability could influence the actual impact of ARs. We also acknowledge that ARs may exert a stronger influence on surface melt in recent, warmer decades or later in spring, when snow and ice are more vulnerable to energy inputs. Therefore, we have incorporated a new discussion section (now Section 6.4, Limitations of the analysis) to better acknowledge the limitations of our seasonally averaged approach and to highlight the potential for greater AR impact in warmer recent years, as follows in Lines 913-919:

"This study is based on seasonal-mean composites from 1980 to 2019, which inherently mask the interannual and intraseasonal variability of AR-induced SEB anomalies. While the analysis discusses the potential seasonal climatological impacts of ARs, their actual impact on sea ice and glacier melt is likely modulated by the background climate state. For example, in warmer recent decades or in late spring when surface snow and ice are more susceptible, the same AR-induced energy anomalies may exert stronger melt impacts than in earlier, colder periods. Furthermore, seasonal averaging may obscure temporal biases within the season—if ARs tend to occur in early spring, their potential to affect surface melt could be diminished despite large seasonal-mean anomalies."

The AR induced anomalies resulted from the limited days that the ARs occurred. Following a simple calculation based on the AR occurrence frequency, the percentages of 10-12% mean approximately 10 days of AR occurrence in individual seasons. We don't have knowledge on how long the AR caused anomalies are maintained and affected. The impacts of the anomalies are likely dependent on the maintenance length. This is considerably complex. The manuscript could give a misunderstanding like that the ARs induced anomalies have likely continuous impacts. Therefore, you have clearly to describe the limitations of composite analysis, with the related discussions.

Response: We thank the Reviewer for raising this important point regarding the persistence of AR-induced SEB anomalies. We agree that the impacts of the anomalies are likely dependent on the maintenance length. We have also included this point in the new discussion section (Section

6.4) to clearly acknowledge the limitations of our composite approach, as follows in Lines 919-913:

"Another limitation stems from the composite approach itself: AR-induced anomalies reflect short-lived perturbations associated with limited AR occurrences (10–12% or ~10 days per season). Our results suggest the potential cumulative impact of these short events, but they should not be interpreted as sustained seasonal effects. The actual influence depends on the persistence and timing of the anomalies, which we do not resolve in this analysis. Future work using event-based or lagged analyses is needed to evaluate the duration and full impact of ARs on sea ice and glacier melt."

The result section is quite long. Thus it needs compactly to rewrite it, particularly 3.2.1, which is recommended to divide into the sections of LW and SW.

Response: We thank the Reviewer for this helpful suggestion. In response to this comment, we have substantially reorganized the Results section during our first-round revision. Specifically, the original Section 3.2.1 has now been divided into three more focused subsections to improve clarity and readability:

- **Section 4.1.1**: Surface downward longwave radiation (LWD)
- **Section 4.1.2**: Net surface longwave radiation (LWN)
- **Section 4.1.3**: Net surface shortwave radiation (SWN)

The discussion section includes a discussion about uncertainty related to two methods in AR production. The difference only represents the impact of the different spatial scales. Despite the two methods were used similar sources of reanalysis data, they resulted in larger differences in AR detection. It additionally needs a comparison with the AR result detected from the different reanalysis data with this result.

Response: We appreciate the Reviewer's thoughtful comment regarding the need to further examine uncertainties associated with reanalysis products. However, extensive literature has shown that the variability in AR characteristics and their associated impacts is far more sensitive to the choice of AR detection methods than to the differences among reanalysis datasets themselves (e.g., Collow et al., 2022; O'Brien et al., 2022; Shields et al., 2022). In polar regions specifically, studies have demonstrated that AR characteristics and impacts are largely consistent across different reanalysis datasets—both in the Arctic (Zhang et al., 2023) and the Antarctic (Wille et al., 2019; Wille et al., 2021).

As noted in Section 2.1 of our manuscript, we selected the 85th percentile IVT-based AR index applied to ERA5 because ERA5 provides higher spatial resolution  $(0.25^{\circ} \times 0.25^{\circ})$  and has been shown to offer improved accuracy for AR detection and tracking compared to the coarser-resolution MERRA-2  $(0.5^{\circ} \times 0.625^{\circ})$ , particularly in polar regions (Zhang et al., 2023). Given this, we believe our current approach using ERA5 offers a more reliable assessment of AR features in the Arctic.

It is helpful if you could provide quantitative assessments on how the AR induced anomalous SEB, for example LWD 20 W/m2, contributes to sea ice melting or glacier.

Response: We appreciate the Reviewer's suggestion to quantitatively assess the contribution of AR-induced SEB anomalies—such as the example of a +20 W/m² LWD anomaly—to sea ice or glacier melting. This is an important point, and we have added the corresponding statement to provide a first-order estimate of the potential melt energy implied by the anomalous SEB. We have added this estimate in Section 4.3 of the revised manuscript to better contextualize the physical implications of the observed SEB anomalies in Lines 698-703:

"To provide context, we note that and additional  $+20 \text{ W m}^{-2}$  sustained over a 10-day AR period over one season corresponds to a cumulative energy input of approximately  $17.3 \text{ MJ m}^{-2}$ . Assuming all of this energy goes into melting ice and using the latent heat of fusion for ice ( $\sim$ 334 kJ kg $^{-1}$ ), this would be sufficient to melt roughly 5.2 cm of ice ( $52 \text{ kg m}^{-2}$  of sea ice). While this is a simplified calculation and does not account for other energy sinks such as sensible heat flux, conduction, or the energy required to melt snow first, it does suggest that AR-induced anomalies can provide meaningful energy for localized melt, particularly near sea ice margins or glacier ablation zones."

The conclusion section seems a repeat of the discussion section. It has compactly to resummarize the major results of this study firstly, then the meaning of the finding and influence, including the lacking and next step.

Response: We thank the Reviewer for this thoughtful suggestion. In response, we have revised the Conclusion section to be more concise and focused. It now summarizes the key findings, highlights their broader implications for Arctic climate, with the lacking and next step detailed stated in the Discussion section.

Based on these comments, the manuscript is needed a major revision.

Response: We appreciate the Reviewer's detailed and constructive feedback. In this second-round revision, we have carefully addressed all comments and incorporated the suggested changes throughout the manuscript. We believe these revisions have substantially improved both the clarity and overall quality of the study.

**References:**

- Collow, A. B. M., Shields, C. A., Guan, B., Kim, S., Lora, J. M., McClenny, E. E., et al. (2022). An Overview of ARTMIP's Tier 2 Reanalysis Intercomparison: Uncertainty in the Detection of Atmospheric Rivers and Their Associated Precipitation. *Journal of Geophysical Research: Atmospheres*, 127(8). https://doi.org/10.1029/2021JD036155
- O'Brien, T. A., Wehner, M. F., Payne, A. E., Shields, C. A., Rutz, J. J., Leung, L. R., et al. (2022). Increases in Future AR Count and Size: Overview of the ARTMIP Tier 2 CMIP5/6 Experiment. *Journal of Geophysical Research: Atmospheres*, 127(6). https://doi.org/10.1029/2021JD036013
- Shields, C. A., Wille, J. D., Marquardt Collow, A. B., Maclennan, M., & Gorodetskaya, I. V. (2022). Evaluating Uncertainty and Modes of Variability for Antarctic Atmospheric Rivers. *Geophysical Research Letters*, 49(16). https://doi.org/10.1029/2022GL099577
- Wille, J. D., Favier, V., Dufour, A., Gorodetskaya, I. V., Turner, J., Agosta, C., & Codron, F. (2019). West Antarctic surface melt triggered by atmospheric rivers. *Nature Geoscience*, *12*(11). https://doi.org/10.1038/s41561-019-0460-1

- Wille, J. D., Favier, V., Gorodetskaya, I. V., Agosta, C., Kittel, C., Beeman, J. C., et al. (2021). Antarctic Atmospheric River Climatology and Precipitation Impacts. *Journal of Geophysical Research: Atmospheres*, 126(8). https://doi.org/10.1029/2020JD033788
- Zhang, C., Tung, W., & Cleveland, W. S. (2023). Climatology and decadal changes of Arctic atmospheric rivers based on ERA5 and MERRA-2. *Environmental Research: Climate*. https://doi.org/10.1088/2752-5295/acdf0f